# Regulation of Notch output dynamics via specific E(spl)-HLH factors during bristle patterning in *Drosophila*

Lydie Couturier[1,2], Khalil Mazouni[1,2], Francis Corson [3] & François Schweisguth[1,2]

The stereotyped arrangement of sensory bristles on the adult fly thorax arises from a self-organized process, in which inhibitory Notch signaling both delimits proneural stripes and singles out sensory organ precursor cells (SOPs). A dynamic balance between proneural factors and Enhancer of split-HLH (E(spl)-HLH) Notch targets underlies patterning, but how this is regulated is unclear. Here, were identify two classes of E(spl)-HLH factors, whose expression both precedes and delimits proneural activity, and is dependent on proneural activity and required for proper SOP spacing within the stripes, respectively. These two classes are partially redundant, since a member of the second class, that is normally cross-repressed by members of the first class, can functionally compensate for their absence. The regulation of specific *E(spl)-HLH* genes by proneural factors amplifies the response to Notch as SOPs are being selected, contributing to patterning dynamics in the notum, and likely operates in other developmental contexts.

---

[1] Department of Developmental and Stem Cell Biology, Institut Pasteur, 75015 Paris, France. [2] CNRS, UMR3738, 75015 Paris, France. [3] Laboratoire de Physique de l'Ecole Normale Supérieure, CNRS, Sorbonne Université, Université Paris Diderot, 75005 Paris, France. Correspondence and requests for materials should be addressed to F.S. (email: fschweis@pasteur.fr)

Pattern formation is a central question in developmental biology. Patterns of differentiated cells that are invariant across individuals can be observed in many species. These can be generated by interpreting a fixed pre-pattern or via self-organization, possibly guided by fixed initial conditions[1,2]. The patterns of sensory organs in *Drosophila* are model systems to study how simple patterns dynamically emerge during development[3]. On the dorsal thorax, or notum, macrochaetae are found at fixed positions and microchaetae are regularly distributed in five dorso-central rows in each hemi-notum (Fig. 1a). As each of these sensory bristles develops from a single sensory organ precursor cell (SOP)[4,5], their pattern in the adult results from the pattern of SOPs in imaginal tissues in late third instar larvae and early pupae. SOPs emerge from groups of cells that express one or more transcriptional activators of the bHLH proneural family, e.g., Achaete (Ac) and Scute (Sc). These factors confer these cells with the ability to become SOPs and these groups of cells are known as proneural clusters[6]. Thus, the position-specific expression of Ac and Sc determine where sensory bristles can develop[7]. Two models have been proposed to explain how proneural clusters develop. In a first model, positional cues govern early proneural activity, and thereby the stereotyped layout of sensory organs whereas inhibitory Notch signaling acts downstream of proneural activity to select SOPs within each proneural cluster. This model applies well for the macrochaetae[8]. In a second model, it is instead Notch activity that negatively defines where proneural activity can emerge, and self-organization involving cell–cell interactions mediated by Notch directs patterning dynamics at the tissue scale (Fig. 1b). In this model, the role of positional cues is limited to defining the initial and/or boundary conditions that guide self-organized Notch dynamics. This model is proposed to apply for the five rows of microchaetae in the dorsal-central notum, which arise from a series of proneural stripes that emerge in a defined sequence[9].

Once proneural clusters or stripes have formed, inhibitory cell–cell interactions mediated by Notch restrict the potential to become an SOP to one or a few cells per cluster (or stripe)[6,10,11]. Notch inhibits the competence to become neural via the E(spl)-HLH family of transcriptional repressors[12–18] which act redundantly to antagonize the activity and expression of Ac and Sc[19–23]. Therefore, adoption of the SOP fate depends on a balance between the activity of Ac and Sc, acting synergistically with Senseless (Sens)[24], and the anti-proneural activity of the E(spl)-HLH proteins[25]. How this balance is dynamically regulated during patterning in the notum is not clear, in part because the dynamic expression of individual E(spl)-HLH factors is not known. While all seven E(spl)-HLH genes are directly regulated by Notch[13,14,26,27], each of these genes has, however, a unique expression pattern[28]. The transcriptional response of the E(spl)-HLH genes to Notch is therefore context-specific, and spatially restricted factors, including Ac and Sc, appear to cooperate with Notch for their regulation in embryos and imaginal tissues[26,27,29–34]. Thus, whether and how different E(spl)-HLH factors contribute to the evolution of the proneural pattern and the emergence of isolated SOPs remains to be studied.

Here, we examine the role of Notch signaling in early stripe patterning and characterize the expression and function of the different E(spl)-HLH factors in the developing notum. Using reporters for each of the seven E(spl)-HLH proteins, we find that a subset of E(spl)-HLH factors are expressed early, prior to the onset of proneural activity, and that additional E(spl)-HLH factors become expressed late, in a proneural-dependent manner. Early-onset factors define where the first stripes emerge while late-onset factors contribute to SOP selection. We propose that the regulation of specific E(spl)-HLH factors by Ac and Sc promotes mutual inhibition through a proneural-dependent increase in the number of E(spl)-HLH genes responding to Notch in cells with intermediate levels of Ac and Sc.

## Results

**Mindbomb1 is required for stripe patterning.** In the early notum, proneural stripes are proposed to be negatively defined by Notch[9]. Specifically, a bimodal gradient of Dl is decoded through trans-activation and cis-inhibition to produce a stereotyped pattern of Notch activity (Fig. 1b, c)[9]. Since Notch signaling blocks the expression of Ac and Sc, only cells with no, or low, Notch activity express Ac and Sc, forming a first set of three proneural stripes (Fig. 1b). Then, as the initial Dl gradient recedes and a new central stripe of Dl emerges, the pattern of Notch activity evolves, allowing for the emergence of two new proneural stripes where Notch activity becomes low (Fig. 1b)[9].

This model predicts that loss of Notch signaling in the whole notum should disrupt stripe patterning. To test this prediction, we studied patterning in *mindbomb1* (*mib1*) mutant pupae. Mib1 is an E3 ubiquitin ligase that is essential for only a subset of Notch signaling events as it acts redundantly with Neuralized (Neur)[35–37]. Neur, however, is not expressed in the early notum and is not required for stripe patterning[9,38], suggesting that Mib1 regulates Notch receptor activation during early stripe patterning. Indeed, a strong loss of the E(spl)m3-HLH (m3) factor (GFP-m3, Fig. 1c, d) and of a m3-GFP transcriptional reporter (m3-GFP, Fig. 1e, f; see below for a description of this reporter) was observed in *mib1* mutant pupae. Also consistent with a loss of Notch activity, a broad domain of GFP-Sc expression was detected at the position of stripes 2–4 in *mib1* mutant pupae (Fig. 1g, h), showing that the patterning of proneural stripes 2–4 requires Mib1-dependent Notch signaling. Moreover, adult *mib1* mutant flies showed not only an increase in bristle density but also no clear pattern of rows (Fig. 1i; see also[36]). Thus, Mib1 is required for the patterning of bristle rows, and also contributes to SOP selection, a process that requires the activity of Neur[35–37]. We conclude that Mib1 regulates the patterning of proneural stripes in early pupae and that Notch acts as a negative template to define where proneural stripes can emerge.

**The E(spl)-C encode the key Notch targets for stripe patterning.** Since the m3 gene is a target of Mib1-dependent Notch signaling, we wondered whether it mediates the negative template activity of Notch. Analysis of flies homozygous for a null m3 allele produced by CRISPR-mediated non-homologous end-joining showed that the m3 gene is dispensable for bristle patterning (Supplementary Fig. 1). Thus, additional Notch targets must act in parallel to m3 to define proneural stripes.

To test whether the m3 gene acts redundantly with other E(spl)-C genes[12–17], large clones of cells homozygous for the E(spl) 32.2 deletion were generated in the notum using the Minute technique. This resulted in early pupal lethality, hence preventing us from studying stripe patterning. We therefore developed an alternative flp-FRT approach to delete in a conditional and tissue-specific manner the E(spl)-C in all, or most, cells of the developing notum. Using genome engineering, we introduced FRT sites on both sides of the E(spl)-C. First, two distal FRT sites, flanking the m7 and m8 genes, and a 3xP3-RFP marker were introduced by CRIPSR-mediated homologous recombination (HR; Fig. 2a). Second, these engineered modifications were recombined with a proximal FRT carried by the PBac{PB}Nf1c00617 element marked by the *white* gene by selecting a recombinant *white*+ RFP+ chromosome. We then used flp-out recombination to produce an E(spl)-C locus flanked by single FRT sites (Fig. 2b; deletion of the m7 and m8 genes did not affect viability, see below). Conditional flp-out of the entire E(spl)-C was then performed over Df(3)P11, a

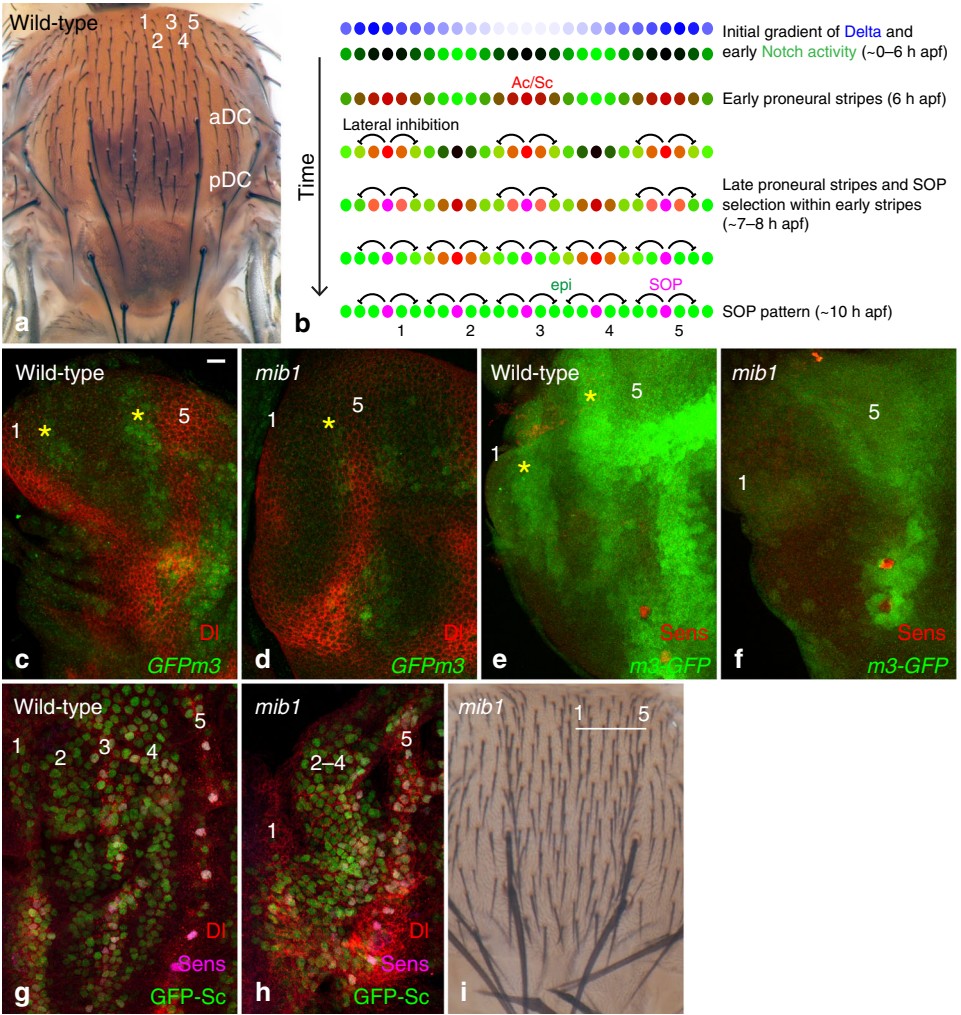

**Fig. 1** *mib1*-dependent Notch activity is required for stripe patterning. **a** A dorsal view of a wild-type adult fly showing the five regular rows of microchaetae (1–5) on each hemi-notum. The anterior and posterior DC macrochaetae (aDC and pDC) are located at the base of row 5. **b** Patterning dynamics in the notum[9]: downstream of a bimodal gradient of Dl, a proneural-independent pattern of Notch activity (green) determines the position of the first proneural stripes (proneural activity, red). As the pattern of Notch activity evolves, two additional proneural stripes emerge and SOPs (magenta) are selected within each stripe. **c**, **d** GFPm3 (GFP, green) is expressed in two stripes of cells (yellow asterisks) flanking cells expressing high levels of Dl (red; located at the position of the future stripes 1 and 5) in 2.5 h apf wild-type nota (**c**). The expression of GFPm3 is strongly reduced in *mib1* mutant pupae (**d**). **e**, **f** A m3-GFP transcriptional reporter (GFP, green) is expressed at high levels in two stripes of cells (yellow asterisks) flanking stripes 1 and 5 in 2.5 h apf wild-type nota (**e**). Expression of the m3-GFP transcriptional reporter is strongly reduced in *mib1* mutant pupae **f** Sens, red, marks cells of the DC macrochaetae along stripe 5. **g**, **h** Five proneural stripes (GFP-Sc, green) are observed in wild-type 7–8 h apf pupae (**g**). Note that the expression of GFP-Sc is already resolved to singled-out SOPs in row 5 (Sens, magenta; Dl, red). In *mib1* mutant pupae (**h**), a broad domain of proneural cells expressing GFP-Sc is observed in the region corresponding to stripes 2–4, whereas stripes 1 and 5 can be identified. **i** Adult *mib1* mutant flies exhibit a disordered array of regularly spaced bristles, but no clear pattern of rows, in the DC region of the notum. Bristle density is increased. Scale bar is 10 μm (**c**: **c–h**). In this and all other figures, representative images of >6 samples (>2 images per sample), from >2 experiments are shown

41 kb deletion that removes all *E(spl)-C* gene but *mδ*[39], to produce notum cells *trans*-heterozygous for a 55 kb *Nf1-m8* deletion over *Df(3)P11*. This resulted in adult flies showing a strong bristle loss in the notum (Fig. 2g, l) and small but properly patterned wings with enlarged veins (Supplementary Fig. 1). These phenotypes are consistent with an efficient tissue-specific knock-out of the *E(spl)-C*. Indeed, the bristle loss appeared to result from a loss of Notch signaling: SOPs failed to single out and neurons were produced at the expense of external sensory cells (Fig. 2e, f, j, k). Thus, flp-out of the *E(spl)-C* is an efficient tissue-specific gene knock-out approach. We therefore used this approach to study the role the *E(spl)-HLH* genes in stripe patterning. Deletion of the *E(spl)-HLH* genes in *E(spl)-C*[FRT3]/*Df(3)P11* larvae disrupted stripe patterning: while control pupae showed a proper pattern of cells expressing GFP-Sc and Senseless (Sens) at 7 h after puparium formation

(apf), nearly all cells accumulated GFP-Sc upon deletion of the *E(spl)-C* (Fig. 2c, d, h, i). We conclude that the E(spl)-HLH factors are the key Notch targets that negatively define the initial stripes of Sc-expressing cells. This implied that *E(spl)-HLH* genes other than the *m3* gene are involved in early stripe patterning.

**A complete set of GFP-tagged E(spl)-HLH reporters**. We therefore next determined the expression dynamics of all E(spl)-HLH factors in the notum. Since the only available antibody, mAb323, recognizes four E(spl)-HLH proteins (mδ, mγ, mβ, and m3)[18], we opted for GFP tagging individual factors. We previously used a functional *E(spl)-C* BAC transgene and GFP tagging to monitor the dynamics of m3[9] and m8[40]. Here, we have tagged the remaining five E(spl)-HLH proteins (Fig. 3a). These

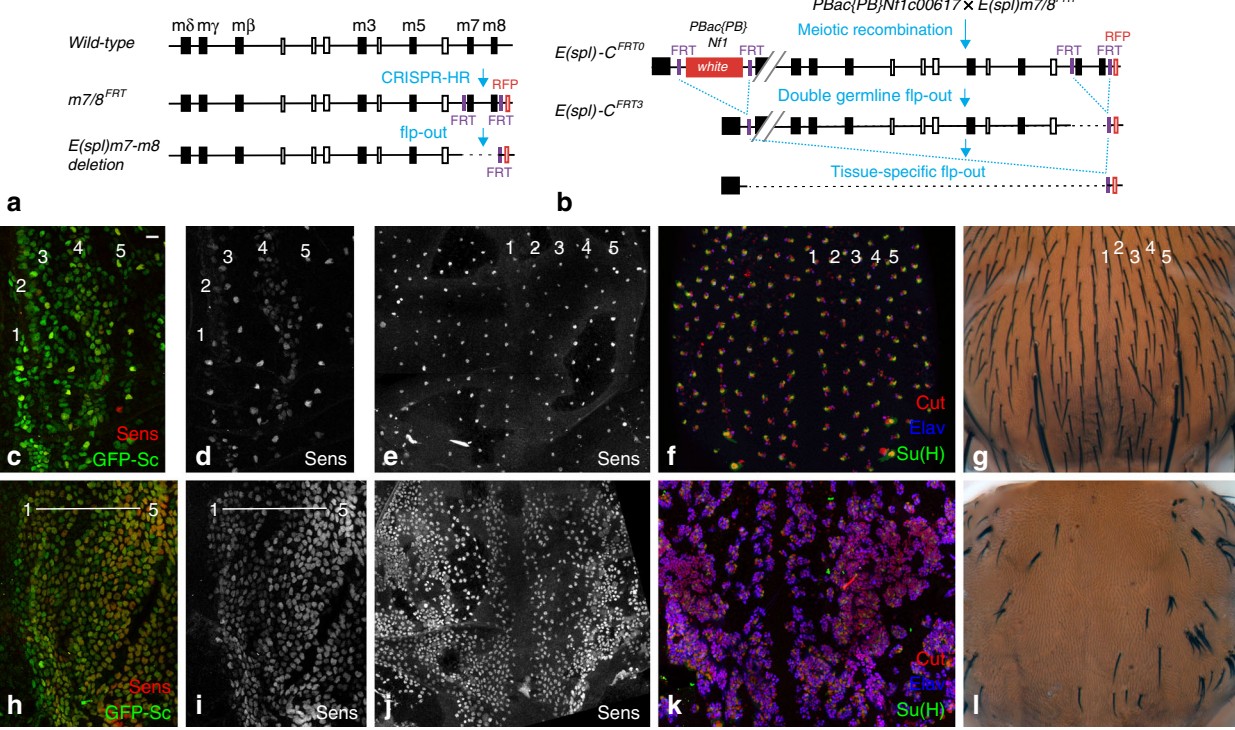

**Fig. 2** the *E(spl)-C* genes mediate the negative template activity of Notch. **a** Two FRT sites flanking the *m7* and *m8* genes were inserted by CRISPR-mediated HR to generate the *m7/8^FRT* chromosome. A small *E(spl)m7-m8* deletion was then generated by flp-out. **b** Tissue-specific knock-out of the *E(spl)-C*. Recombination between a FRT located within a PiggyBac element inserted into the *Nf1* gene (marked by *white+*) and the FRT sites of the *m7/8^FRT* chromosome (marked by *RFP*) produced the *E(spl)-C^FRT0* chromosome. Flp-mediated deletion of the *white* marker, leaving a single proximal FRT, and of the *m7* and *m8* genes, leaving a single distal FRT, produced the *E(spl)-C)^FRT3* chromosome which was used for tissue-specific flp-out to produce a 55 kb *E(spl)-C* deletion. **c–l** Sensory organ development was studied in control (**c–g**; +/*E(spl)-C^FRT3 ap^ts > flp* pupae) and deletion nota (**h–l**; *Df(3)P11/ E(spl)-C^FRT3 ap^ts > flp* pupae). At 7 h apf, GFP-Sc (green) and Sens (red) were expressed in a pattern of five proneural stripes in control pupae (**c**, **d**), similarly as in wild-type pupae (Fig. 1g) whereas nearly all notum cells expressed GFP-Sc and Sens upon conditional knock-out (**h**, **i**). Thus, the *E(spl)-C* genes are required for the patterning of the proneural stripes. Regular rows of SOPs (**e**, 15 h apf) and sensory bristles (**f**, 24 h apf; Elav, blue, is a neuronal marker; Cut, red, marks all sensory cells; Su(H), green, marks the socket cells; g, adult notum) were seen in control flies. In contrast, too many SOPs (j, 15 h apf) produced mostly positive neurons (k, 24 h apf), hence leading to a strong bristle loss phenotype (**l**) upon notum-specific deletion of the *E(spl)-C*. These *Notch*-like phenotypes indicate that the *E(spl)-C* genes are key Notch targets for the formation of sensory organs. Scale bar is 10 μm (**c**: **c–l**)

GFP-tagged proteins were found to accumulate in a pattern similar to those reported earlier for the corresponding RNAs in wing imaginal discs, suggesting that these BAC transgenes faithfully report the E(spl)-HLH protein expression patterns (Fig. 3b–e and h–j). We also generated a GFP-tagged version of m3 at the locus using CRISPR-mediated HR. GFPm3 produced from the endogenous locus and from the BAC transgene exhibited identical expression patterns (Fig. 3a, f), further validating our BAC transgene approach. Consistent with the notion of context-specific expression[26,27,29,31], these reporters, and more generally expression of the *E(spl)-HLH* genes, do not necessarily reflect the strength of Notch activity, and their individual or combined expression do not necessarily match the pattern of Notch signaling, which might be revealed using a synthetic NICD-responsive reporter[41] (Supplementary Fig. 2).

Proteins of the E(spl)/HES family are usually thought to be unstable[42]. To test whether the E(spl)-HLH moiety confers instability to the GFP fusion protein, we generated a BAC transcriptional reporter for the *m3* gene by replacing its open reading frame (ORF) by GFP, keeping the 5' and 3' untranslated regions of the mRNA intact (Fig. 3a). A stronger GFP signal and a broader pattern of expression were seen with this m3-GFP transcriptional reporter compared to the GFPm3 fusion, indicating that GFPm3 turns over more rapidly than GFP (Fig. 3g and Supplementary Fig. 2). Together, our data indicate that these

short-lived reporters can provide insights into the dynamics of the endogenous E(spl)-HLH proteins.

**m3 and mβ mediate the negative template activity of Notch.** Using this set of GFP-tagged E(spl)-HLH factors, we first confirmed that GFPm3 is expressed in the notum at 2.5 h apf in cells flanking the Dl stripes (Fig. 4c)[9] and found that GFPmβ, but no other E(spl)-HLH, was expressed like GFPm3 (Fig. 4b). GFPmδ and GFPm7 were detected in dorsocentral (DC) cells (Fig. 4a, d), suggesting that these two genes respond to the inhibitory signal produced by the DC macrochaetes. Thus, m3 and mβ appear to be the only E(spl)-HLH factors acting downstream of Notch to negatively define the first proneural stripes[9]. We refer here to m3 and mβ as the early-onset E(spl)-HLH factors.

To test whether mβ acts redundantly with m3 for stripe patterning, we generated a *m3^CR1 mβ^CR1* double mutant chromosome (Fig. 5a and Supplementary Fig. 1). Surprisingly, we found that homozygous *m3^CR1 mβ^CR1* and *trans*-heterozygous *m3^CR1 mβ^CR1 / Df(3)X10* mutant flies were viable and showed no bristle patterning defects (Fig. 5b and Supplementary Fig. 1; *Df(3)X10* removes the *mδ-m3* genes[39]). Moreover, the first three proneural stripes appeared to be properly defined in 7 h apf pupae in the absence of both m3 and mβ (Fig. 5c, d). We therefore suggest that E(spl)-HLH factors other than mβ and m3 must act downstream

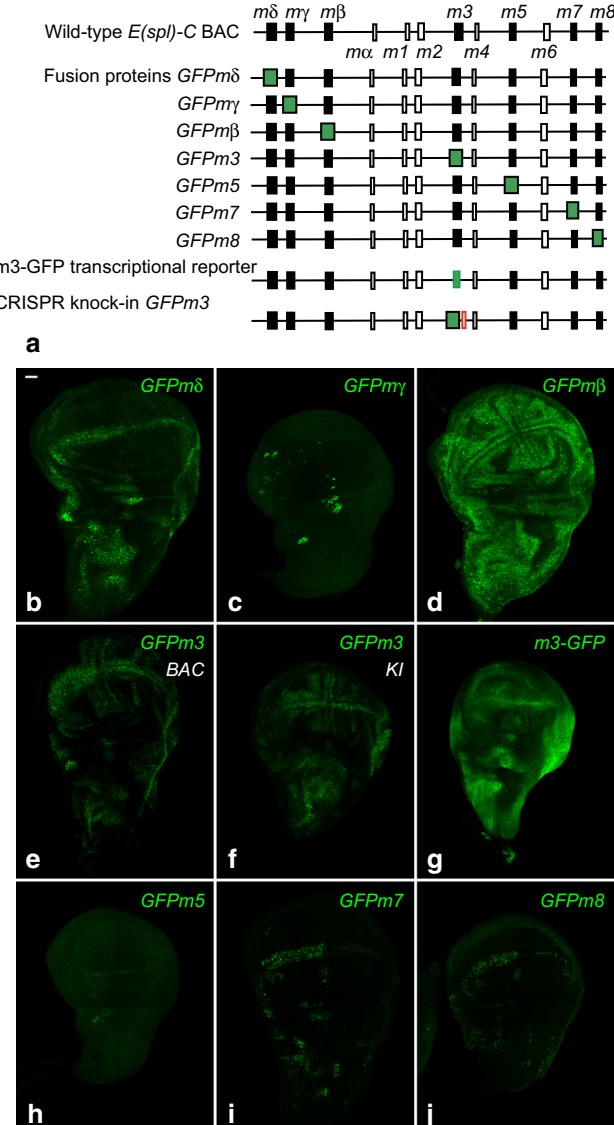

**Fig. 3** a complete set of E(spl)-HLH reporters. **a** Genomic structure of the *E (spl)-C* BAC[59]. The seven *E(spl)-HLH* genes are shown in black, other genes appear as open boxes. The complete series of BAC transgenes encoding GFP-tagged fusion proteins, as well as the BAC transgene encoding a transcriptional reporter for the *m3* gene, *m3-GFP*, are diagrammatically shown. A CRISPR knock-in locus with GFP-tagged m3 and a FRT site 3' to the *m3* gene is also shown. **b–j** Expression pattern of these E(spl)-HLH reporters (GFP, green) in third instar wing imaginal discs. Both the *GFPm3* BAC transgene (**e**) and CRISPR knock-in line (**f**) are shown. Scale bar is 15 μm (**b**: **b–j**)

of Notch to pattern the proneural stripes, at least in the *mβ m3* double mutant.

Having shown that the key Notch targets appear to reside within the *E(spl)-C* and that m3 and mβ are the only two early-onset factors, we wondered whether one (or more) *E(spl)-HLH* genes might be de-repressed in *m3 mβ* double mutant pupae, thereby compensating for the loss of m3 and mβ. To test this, we used Fluorescent In Situ Hybridization (FISH) to examine the expression of the *mδ, mγ, m5, m7,* and *m8* genes. We first confirmed that these genes are not detectably expressed in wild-type 0–2 h apf nota (Fig. 5e, f, i, j). However, two stripes of ectopic *mδ* expression were detected in the DC region of *mβ m3* mutants (Fig. 5g, h), in a pattern similar to those of m3 in wild-type pupae.

Thus, loss of the m3 and mβ factors appeared to result in a specific de-repression of the *mδ* gene. In contrast, no significant upregulation was detected for the *mγ, m5, m7,* and *m8* genes in the notum of mutant pupae (Fig. 5k, l; note, however, that upregulation was observed distally). We therefore conclude that m3 and/or mβ negatively cross-regulate the expression of the *mδ* gene in the notum. This de-repression of *mδ* may account for stripe patterning in *m3 mβ* double mutant pupae.

This interpretation predicts that the combined loss of mδ, mβ and m3 should result in stripe patterning defects. To test this, we generated large clones of cells mutant for the *E(spl)mδ-m6* deletion, which removes these three *E(spl)-HLH* genes, along with the *mγ* and *m5* genes, but retains the *m7* and *m8* genes[43]. While the initial gradient of Dl was normal in these mutant nota (Fig. 5m), this spatial information was not properly decoded through Notch: GFP-Sc and Sens were broadly expressed where stripes 2–4 would normally emerge at 7–8 h apf (Fig. 5n, o). In addition, emerging SOPs with high GFP-Sc and Sens levels form a dense and regular two-dimensional array (Fig. 5n, o) instead of regular rows (Fig. 2d). Also, adult flies showed an increased number of regularly spaced bristles with no discernable pattern of rows (Fig. 5p). These results support the notion that a combined loss of mδ, mβ and m3 produces stripe patterning defects. Since mδ, mγ, and m5 are not detectably expressed in the notum region of wild-type pupae before 6 h apf, m3 and mβ are likely to be the only E(spl)-HLH factors mediating the negative template activity of Notch. In their absence, mδ is ectopically expressed and can functionally compensate for their loss. We therefore suggest that m3 and mβ are the key Notch targets for proneural stripe patterning.

**Expression of E(spl)-HLH factors during stripe resolution**. Having identified the key Notch targets for early stripe patterning, we then characterized the key Notch targets for stripe resolution and SOP selection. At 7–8 h apf, all seven E(spl)-HLH factors were detectably expressed, albeit at different levels and in distinct patterns (Fig. 6a–g). First, GFPmβ appeared to be expressed similarly as GFPm3[9], on the sides of the proneural stripes 1, 3, and 5 (Fig. 6c, d; Sens was used here as a stripe marker). GFPm3, however, appeared to be expressed at higher levels than GFPmβ. Second, GFPmδ, GFPm7, and GFPm8 appeared to be co-expressed in a subset of cells within proneural stripes (Fig. 6a, f, g and Supplementary Fig. 3). Third, GFPmγ and GFPm5 were expressed in only few cells at the position of proneural stripe 5 (Fig. 6b, e). Thus, five of the seven E(spl)-HLH factors are dynamically expressed at significant levels during bristle patterning.

Using a BAC-encoded RFP-Ac to monitor the expression of Ac[9], we found that GFPm3 accumulated only in cells with only low (or no) Ac, which were found on the sides of the proneural stripes whereas Ac-positive cells at the center of the proneural stripes showed very low GFPm3 (Fig. 6h–l). In contrast, GFPm7 and GFPm8 were detected in proneural stripe cells with low/intermediate levels of Ac (Fig. 6m–v). Nevertheless, cells with high levels of Ac and Sens, which correspond to emerging SOPs, had very low levels of GFPm7 and GFPm8. Thus, two classes of E (spl)-HLH factors can be defined based on their patterns of expression during bristle patterning. A first class of early-onset factors, comprising m3 and mβ, is characterized by its early expression, prior to the onset of proneural gene expression, and a limited overlap with proneural activity. A second class of late-onset factors, that comprises mδ, m7, and m8, is characterized by their late expression, detected once proneural factors become expressed, and their partial expression overlap with the proneural factors.

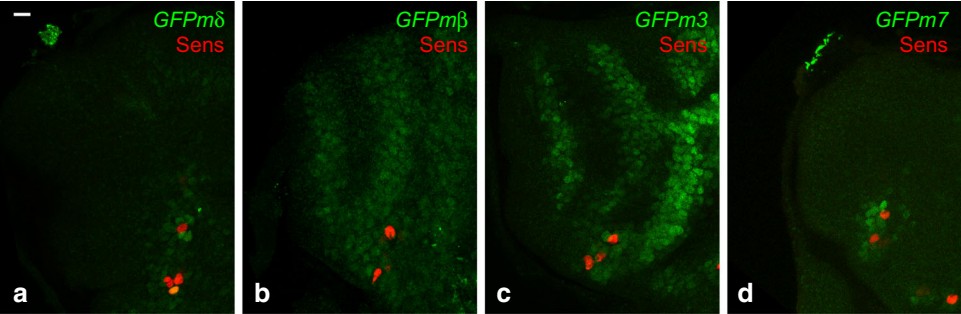

**Fig. 4** *m3* and *mβ* are expressed early in response to the bimodal gradient of Dl. GFPmδ (**a**) GFPmβ (**b**), GFPm3 (**c**), and GFPm7 (**d**) reporters (GFP, green) are detected at 2.5 h apf. GFPmβ is expressed in stripes like GFPm3. GFPmδ, and GFPm7 are weakly expressed in the non-selected cells from the DC proneural cluster. Sens (red) marked sensory cells of the future DC macrochaetae, used here as a landmark position for the future proneural stripe 5. Scale bar is 10 μm (**a**: **a**–**d**)

**Ac and Sc regulate late-onset *E(spl)*-HLH gene expression.** Previous studies have shown that the *m7* and *m8* genes are regulated by Ac and Sc in wing imaginal discs[26,27,33], suggesting that their expression in cells with intermediate Ac and Sc levels might involve a dual input from Notch and Ac (or Sc). In contrast, the *m3* and *mβ* genes are likely to respond to a simple Notch input since they are expressed early, prior to the onset of Ac and Sc expression, and in cells with low proneural activity thereafter. To test the role of Ac and Sc in E(spl)-HLH gene expression in the notum, we studied the expression of GFPmβ, GFPm3, GFPm7 and GFPm8 in *sc10-1* mutant pupae, which lack both Ac and Sc. While the early patterns of GFPm3 and GFPmβ expression were not affected by the loss of *ac* and *sc* activities, the expression of GFPm7 and GFPm8 was lost in *sc10-1* mutant pupae (Fig. 7a–d and Supplementary Fig. 4). We conclude that proneural factors regulate the expression of the late-onset *E(spl)*-HLH genes. Consistent with this, once SOPs are specified and Ac and Sc levels have declined, cells close to SOPs no longer expressed GFPm7 and GFPm8 but still accumulated GFPm3 in response to the Dl signal sent by SOPs (Supplementary Fig. 4). In addition, GFPm7, but not GFPm3, was detected in *Dl Ser* mutant cells (Fig. 7e–h), indicating that the *m7* gene can be activated in the absence of Notch signaling, presumably by Ac, Sc, and Sens, which accumulate at high levels in mutant cells. This is consistent with earlier data showing that disrupting the regulation by Notch of the *m8* gene restricted its expression to cells with high proneural activity[13]. Thus, differences in expression dynamics amongst *E(spl)*-HLH genes appear to result from gene-specific regulation by Ac and Sc in the notum.

**m7 and m8 regulate the spacing of SOPs within stripes.** Our data above suggested that m3 and mβ contribute to the early exclusion of cells with low proneural activity on the edges of the proneural stripes, whereas mδ, m7, and m8 contribute to the late exclusion of cells with high proneural activity at the center of the stripes. If m7 and m8 participate to the late exclusion of cells at the center of stripes, we predict that an excess number of bristles would be produced without affecting the pattern of rows in their absence. To test this, we produced a small *E(spl)m7-m8* deletion through flp-mediated recombination of the two FRT sites flanking the *m7* and *m8* genes (Fig. 2a). Flies homozygous for this deletion, or carrying this deletion over a large *E(spl)-C* deletion, were viable, exhibited a normal pattern of five rows but showed a significant increase in microchaetae number (Fig. 8a): 150+/−6 (mean + /− standard error of the mean; $n = 4$) and 184+/−11 ($n = 4$) bristles were scored in *E(spl)m7-m8/Df(3)P11* male and female flies, respectively, as compared with 114+/−5 ($n = 5$; $p <$

0.0001, *t*-test) and 132+/−5 ($n = 7$; $p < 0.001$, *t*-test) in wild-type flies. We therefore conclude that the proneural-dependent expression of the *m7* and *m8* genes is important for the proper spacing of SOPs within each stripe.

This increase in bristle density is, however, relatively moderate, possibly because some E(spl)-HLH factors acting redundantly with m7 and m8 for SOP selection are derepressed in the *E(spl) m7-m8* deletion mutant. Consistent with this, we found that the *m5* gene was expressed at higher levels in *E(spl)m7-m8* mutant relative to wild-type nota at 8 h apf (Fig. 8b, c). Thus, our deletion analysis likely underestimates the actual contribution of m7 and m8 in preventing proneural-expressing cells from adopting the SOP fate.

Finally, we confirmed that the deletion of the *E(spl)-C* in clones of cells homozygous for the *E(spl)32.2* deletion resulted in tufted bristles with no intervening epidermal cells[25] whereas a deletion removing all *E(spl)-C* genes but *m7* and *m8* led to regularly spaced bristles, albeit at increased density (Fig. 8d, e; see also Fig. 5p). This showed that the m7 and m8 factors are sufficient for the singling out of SOPs. Together, our results strongly suggest that the proneural-dependent expression of the late-onset E(spl)-HLH factors regulates the proper spacing of sensory bristles within each proneural stripe.

## Discussion

An early, widespread and evolutionarily conserved response of the genome to Notch activation is the CSL-dependent transcription of the *HES* family genes. In *Drosophila*, the seven *HES* family genes encoded by the *E(spl)-C* act redundantly[16] and the relative contribution of individual E(spl)-HLH factors to the overall Notch output has remained unclear. Here, we generated a complete set of GFP-tagged reporters and studied the expression dynamics of all E(spl)-HLH factors in the developing notum. We found that m3 and mβ are expressed early in response to Mib1-dependent Notch signaling and that additional E(spl)-HLH factors, notably mδ, m7, and m8, become expressed later, once proneural stripes are established and in a proneural-dependent manner. Early-onset factors appeared to mediate the negative template activity of Notch for early stripe patterning, while late-onset factors, notably m7 and m8, are essential to reach sufficient Notch signaling output for the proper spacing of sensory bristles. Thus, different E(spl)-HLH factors contribute at distinct steps of this patterning process and the regulation of specific *E(spl)*-HLH genes by Ac and Sc plays an important role by raising the level of E(spl)-HLH activity, produced in response to Notch, in groups of cells progressing towards the SOP fate. This upregulation is transient, and once proneural stripes have resolved, the same

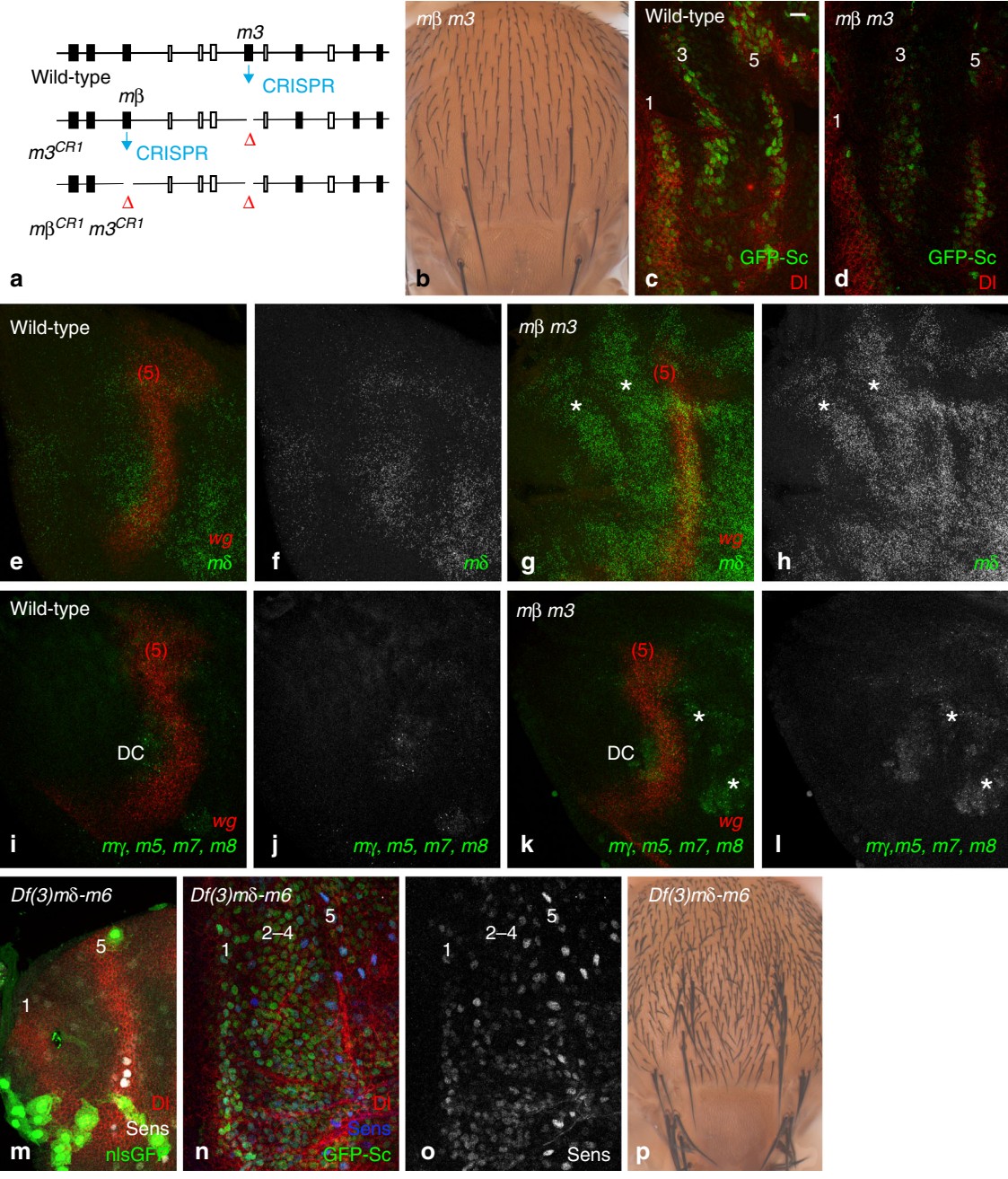

**Fig. 5** *E(spl)-HLH* genes are required for stripe patterning. **a** A two-step CRISPR-based deletion of the *m3* and *mβ* ORFs to rpoduce a double mutant chromosome. **b** Dorsal thorax of an adult *m3 mβ / Df(3)X10* fly showing no bristle patterning defect. The X10 deletion removes the *mδ-m3* genes and extends beyond *mδ*[39]. **c**, **d** The early GFP-Sc proneural stripe pattern (GFP, green; Dl, red) is seen in both wild-type and *m3 mβ* mutant pupae at 7 h apf. **e–h** *mδ* mRNAs (smiFISH probe, green) were not detected in the notum region of wild-type 0–2 h apf pupae (**e**, **f**; *wg* probe, red in **e**; the *wg* stripe, indicated as (5), overlapped with proneural stripe 5). In contrast, *mδ* mRNAs were detected in the notum of *m3 mβ* pupae in a pattern similar to GFPm3 in early wild-type pupae (asterisks in **g**, **h**; see also Fig. 4c). Higher accumulation of *mδ* was also detected in more distal area, indicative of negative regulation of *mδ* by m3 and/or mβ. **i–l** The *mγ, m5, m7, and m8* transcripts (mixed smiFISH probe, green) were not detected in the notum region of wild-type and *m3 mβ* mutant pupae at 0–2 h apf (*wg*, red). Higher levels of *mγ, m5, m7,* and/or *m8* transcripts were detected in the DC cluster and in distal parts of the notum (asterisks in **k**, **l**). **m** Nota comprising only cells homozygous for the *Df(3 L)mδ-m6* deletion, marked by the loss of nlsGFP (green) were obtained using the *Minute* technique (large green cells are non-epithelial macrophages). Dl expression (red; Sens, white) remained unaffected at 2.5 h apf. **n**, **o** Deletion of the the *mδ-m6* genes resulted in defective stripe patterning at 7–8 h apf. A broad domain of GFP-Sc expression (green, **n**) was observed over the stripes 2–4 region, whereas stripes 1 and 5 could be identified (Sens, **o**). This phenotype is similar to the *mib1* mutant phenotype (Fig. 1h). **p** Adult flies harboring *Df(3 L) mδ-m6* mutant cells in the notum show a strongly disordered array of spaced bristles with no pattern of rows but increased bristle density. Scale bar is 10 μm (**c**: **c–o**)

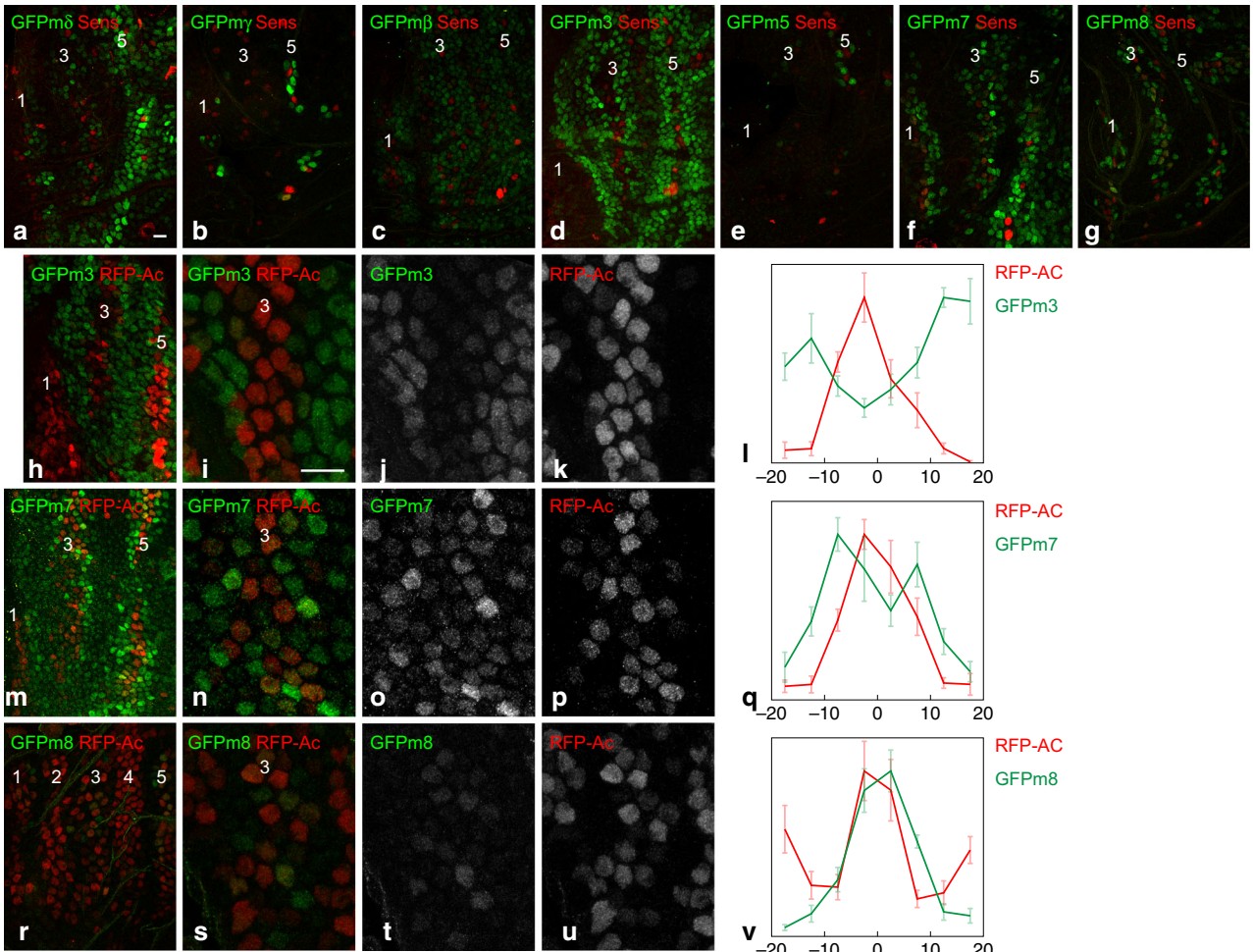

**Fig. 6** Early-onset and late-onset E(spl)-HLH factors exhibit distinct accumulation patterns. **a–g** Expression of GFP-tagged E(spl)-HLH factors (GFP, green) at 8 h apf. The early-onset factors mβ (**c**) and m3 (**d**) appeared to accumulate on the sides of the proneural stripes (Sens, red), whereas the late-onset factors mδ (**a**), m7 (**f**), m8 (**g**) and, to a lesser extent mγ (**b**) and m5 (**e**), appeared to be intermingled with the Sens-positive proneural stripe cells. **h–l** GFPm3 (green) accumulated in cells with very low Ac levels (red), flanking the proneural stripe 3, as shown in the high magnification views of stripe 3 at ~7.5 h apf in **i–k**, and the corresponding quantification of the GFP and RFP nuclear signals in l (n = 5; the mean + / − standard error of normalized intensities is plotted as a function of the distance, in μm, relative to the center of the stripe). **m–v** GFPm7 (green; **m–q**; ~7.5 h apf) and GFPm8 (green; **r–v**; ~8 h apf) accumulated in proneural stripe 3 cells with low/intermediate levels of Ac (red), but not in presumptive SOPs that have high levels of Ac. Note that fewer cells express GFPm8 at the onset of proneural stripe 3, suggesting that GFPm7 may be expressed slightly earlier than GFPm8. High magnification views (**m–p** and **s–u**) and quantifications (**q**, n = 7; **v**, n = 9) as in panels **i–l**. Scale bars are 10 μm (**a**: **a–h**, **m**, **r**; **i**: **i–k**, **n–p**, **s–u**)

early-onset factors that mediate stripe patterning are expressed in non-SOP cells to lock down their fate.

The regulation of the *E(spl)-C* genes by a dual Notch/proneural input has been well studied before[26,27,30,32–34]. This regulation was previously interpreted to suggest that Ac and Sc set up the initial conditions for a regulatory feed-back loop operating between proneural cluster cells[30,33]. In this model, Ac and Sc initiate the conditions for both signaling, via the regulation of the *Dl* and *neur* genes, and responding to Notch, via the regulation of the *E(spl)-C* genes. Our detailed analysis of the expression, regulation and function of the E(spl)-HLH factors did not support this model. Indeed, the proneural-independent expression of m3 and mβ, downstream of Mib1-dependent Notch signaling, appeared to provide the initial conditions for patterning. We propose here a different model, whereby the proneural-dependent regulation of specific E(spl)-HLH factors serves to modulate mutual inhibition within the proneural stripes during SOP selection, and thereby to shape the dynamics of patterning. Cells with intermediate levels of Ac and Sc, because they activate an

increasing number of Notch-responsive *E(spl)-HLH* genes, cannot evade mutual inhibition as readily as they would otherwise. Such a regulatory logic may favor the robust emergence of regularly spaced sensory organs.

A role for the modulation of mutual inhibition was anticipated by an abstract mathematical model that recapitulates the temporal and spatial dynamics of fate patterning in the notum[9], and our findings suggest a molecular basis for several features of the model's dynamics. In brief, this simple model represents the state of each cell by a single variable, *u*. This variable varies in time as a function of an inhibitory signal, *s*, representing the level of Notch ligands to which the cell is exposed, produced by other cells according to their own state *u*. Cells in the model have bistable dynamics, tending to one of two stable cell states, a high *u*/low *s* state (SOP fate; high proneural activity and low inhibitory signal) and a low *u*/high *s* state (non-SOP fate; low proneural activity and high inhibitory signal; Fig. 9a). Given appropriate initial conditions, the model recapitulates the sequential emergence of proneural stripes and their resolution into SOP rows[9]. In our

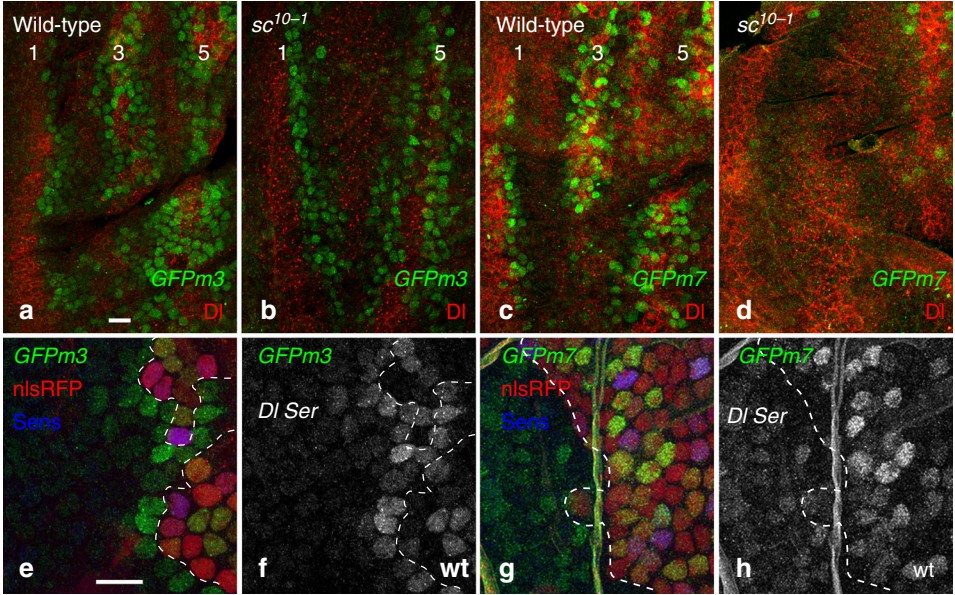

**Fig. 7** Regulation of the late onset *E(spl)-HLH* genes by Ac and Sc. **a–d** Pattern of expression of the GFPm3 (**a**, **b**) and GFPm7 (**c**, **d**) reporters (GFP, green; Dl, red) in the notum of wild-type (**a**, **c**) and *sc^10–1^* (**b**, **d**) 7–8 h apf pupae. While the expression of GFPm3 did not depend on Ac and Sc (**a**, **b**), GFPm7 was not expressed in the absence of Ac and Sc (**c**, **d**) with the exception of a few anterior cells along stripe 5 (**d**). Note that the expression of Dl in stripe 3 did not depend on Ac and Sc[64]. **e–h** GFPm3 (GFP, green in **e**, **f**) was not expressed by *Dl^rev10^ Ser^RX10^* mutant cells (clone border outlined by a white dashed line; non-mutant cells express nlsRFP, red; Sens, blue) with the notable exception of the mutant receiving cells that are in direct contact with the wild-type signal sending cells (**e**, **f**). In contrast, expression of GFPm7 (GFP, green in **g**, **h**) was observed in mutant cells, indicating that expression of the *m7* gene did not strictly depend on ligand-dependent Notch signaling. Clone borders (dotted line) and genotypes (*Dl Ser* vs. wild-type, wt) are indicated. Scale bars are 10 μm (**a**: **a–d**; **e**: **e–h**)

simulations of this mathematical model, the balance between activation and inhibition is such that cells located at the center of the proneural stripes and progressing towards the SOP fate show increasing levels of both proneural activity and inhibitory signal (that is, of *u* and *s*) as seen in the time courses replotted from Fig. 4d in ref. [9] (Fig. 9b–d). Interestingly, these simulations predicted that, once a proneural stripe emerges, the inhibitory signal is strongest in cells at the center of the stripe. This prediction was, however, not verified using GFPm3 as a Notch activity reporter:[9] high levels of GFPm3 were observed in cells flanking the proneural stripes, not at the center of the stripes where SOPs develop[9]. While cis-inhibition of Notch by Dl could account for a discrepancy between the inhibitory signal in the model (representing ligand levels) and Notch activity itself, our detailed analysis of E(spl)-HLH expression suggests a different explanation: this particular Notch target, m3, does not fully reflect the in vivo activity of Notch, and other E(spl)-HLH factors, notably mδ, m7, and m8, contribute to the Notch output, particularly at the center of the proneural stripes. The contribution of cis-inhibition, if any, may thus be limited to emergent SOPs, which exhibit the highest levels of proneural factors and low levels of all E(spl)-HLH factors. The model further tied a gradual narrowing of neural competence, as observed experimentally, to a progressive increase in the strength of mutual inhibition. While this may result from a modulation of ligand activity, e.g., by Neur, in signal-sending cells[9], our findings identify another contribution, from the regulation of Notch target expression in receiving cells.

Our analysis also raises additional questions about how Notch and Ac/Sc regulate gene expression dynamics in the notum. In particular, two observations deserve consideration. First, how is it that the late-onset genes (*mδ*, *m7*, *m8* etc.) do not respond to Notch signaling prior to the proneural onset? Second, why are m3 and mβ are not maximally expressed at the center of the stripes, where inhibitory signaling is predicted to be maximal? Several models can be proposed. In a first accessibility model (Fig. 9e),

only a subset of the CSL-binding sites is accessible for binding by CSL/NICD complexes in the absence of Ac and Sc, so that only a fraction of the *E(spl)-HLH* genes, i.e., *mβ* and *m3*, is activated by Notch; upon expression of Ac and Sc, additional binding sites become accessible, possibly through a change in chromatin structure induced by Ac and Sc, resulting in the expression of additional *E(spl)-HLH* genes. In support of this model, the proneural factor Ascl1 was shown to bind both closed and open chromatin in mouse neural progenitors, and binding to closed chromatin appeared to promote accessibility[44]. In a second cooperativity model (Fig. 9f), early-onset genes contain high-affinity CSL binding sites, whereas late-onset genes have low-affinity CSL binding sites, so that only early-onset genes, i.e., *m3* and *mβ*, respond to a Notch-only input; but the low-affinity CSL binding sites of the late-onset genes would be located close to E-boxes, such that Ac and Sc promote cooperative binding, hence gene expression. While these two models could explain the temporal sequence of gene activation, they do not, however, explain why *m3* and *mβ* are not maximally expressed at the center of the stripes (see below). Two possible mechanisms might account for this observation. One first possibility is that the nuclear concentration of NICD is limiting such that not all CSL binding sites cannot all be occupied, even when Notch signaling is maximal. If so, binding sites would effectively compete for the binding of CSL/NICD. If CSL/NICD complexes preferentially bind the regulatory sites of late-onset genes, this should then result in lower levels of m3 and mβ expression when and where these sites become accessible (accessibility model) or bound by Ac and Sc (cooperativity model). A second possibility is that the expression of the *m3* and *mβ* genes is inhibited by late-onset E(spl)-HLH factors. Further studies will address these different models.

Auto-repression and cross-repression is seen within the *HES* gene family in vertebrates[45,46]. By contrast, self/cross-inhibition by E(spl)-HLH factors had not been observed in *Drosophila*, prior to this study. Here, we showed that the *mδ* gene is de-repressed in

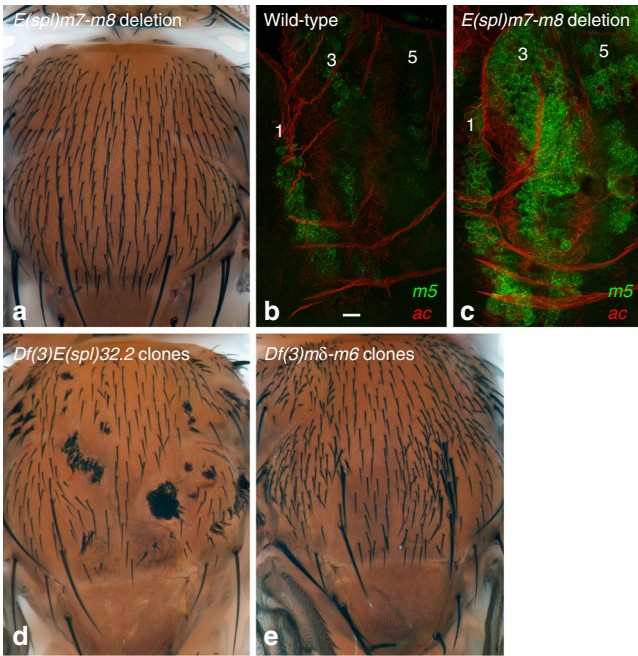

**Fig. 8** The *m7* and *m8* genes contribute to SOP selection. **a** Dorsal thorax of an *E(spl)m7-m8/Df(3)P11* fly, lacking the *m7* and *m8* genes and carrying one copy of the *mγ-m5* genes. An increase in bristle density (+35%) was observed. The *m7* and *m8* genes are required for the proper spacing of SOPs within a stripe but are dispensable for stripe patterning. **b**, **c** The expression of the *m5* gene (smiFISH probe, green) was significantly upregulated in proneural stripes 1,3 and 5, at 8 h apf, in *E(spl)m7-m8* mutant pupae (**c**) relative to wild-type controls (**b**; *ac* probe, red). **d**, **e** Adult flies with clones of cells homozygous for the *E(spl)32.2* deletion, i.e., lacking all *E (spl)-C* genes, show patches of tufted bristles (**d**), whereas clones of *Df(3L) mδ-m6* mutant cells display increased bristle density (**e**). This indicates that the *m7* and *m8* genes are sufficient for the singling out of SOPs. Scale bar is 10 μm (**b**: **b**, **c**)

*mβ m3* double mutant pupae, and ectopic expression in mutant nota appeared strikingly similar to those of the *mβ* and *m3* genes in wild-type pupae. Our finding may help resolve a long-held paradox which is that E(spl)-HLH genes have very distinct expression yet are functionally redundant. Obviously, ectopic expression of one (or several) factor upon loss of one (or several) others would account for redundancy despite specificity in expression.

Our analysis provides no evidence for functional specificity at the molecular level amongst the E(spl)-HLH factors. For instance, m3 and mβ appeared to be functionally replaced by mδ. Similarly, SOP selection could be achieved, at least to some extent, by m7 and m8 alone, as well as by the other five factors, at the exclusion of m7 and m8. Thus, non-overlapping sets of E(spl)-HLH factors can provide a proper Notch output for stripe patterning and SOP selection. These observations, together with earlier findings[12], indicate that the E(spl)-HLH factors have very similar molecular activities. Thus, the proneural-dependent expression of additional E(spl)-HLH factors may simply result in a global increase of an anti-proneural activity that would be provided by any of these factors. Therefore, describing Notch output dynamics and understanding its regulatory logic in a given context may require the analysis of all E(spl)-HLH factors that collectively contribute to this output. The tools generated here will help achieving this.

The patterning logic uncovered here may be of general relevance. While earlier studies viewed SOP selection in larvae as a multi-step process of proneural cluster resolution[33,47], our data suggest that the progressive transition from mutual inhibition

among proneural cells to lateral inhibition from SOPs is dynamically shaped by the regulation of a specific subset of E(spl)-HLH factors by Ac and Sc, with no need for specific mechanisms to restrict competence to a subgroup of proneural cells. In the adult fly gut, a subset of *E(spl)-HLH* genes, including *mδ*, *m7*, and *m8*, are also regulated by Sc and this regulation may be functionally relevant as Sc is a key cell fate regulator within the Intestinal Stem Cell (ISC) lineage[48,49]. Interestingly, ISCs that are in a low Sc state may respond to Notch via mβ and m3 factors, which may suffice to remain in a low Sc state, whereas ISCs that are in a high Sc state may revert to a low Sc state in response to a stronger Notch output that would be produced in part through the mδ, m7, and m8 factors[48,50]. Thus, the regulatory logic unraveled by our study may regulate cell fate within the ISC lineage.

The regulation of the *HES* family genes by a dual Notch/proneural input is evolutionarily conserved. In *Xenopus*, an enhancer integrates the Notch and proneural inputs to regulate the expression of two Hes5-like genes[51]. In the vertebrate brain, Ascl1 regulates the expression of *Hey1*[52], a direct Notch3 target[53]. In this context, it is interesting to note that neural stem cells cycle between a quiescence state, regulated by high Notch3 activity[54,55] and an activated state, also involving Notch3 but in the context of Ascl1 expression[56,57], a combination that might lead to expression of different *HES* family genes, as reported in[53,58]. Actually, in numerous contexts in development, Notch receptor activation intersects with the expression of proneural transcription factors. Thus, the regulatory logic uncovered here for the patterning of sensory organs, involving a proneural-dependent increase in *E (spl)-HLH* gene expression as a mean to upregulate the Notch output, may similarly operate in mammals.

## Methods

**BAC transgenes and CRISPR-based genome engineering**. The BAC transgenes encoding the GFP-tagged E(spl)-HLH proteins were generated from an attB-P [acman]-Ap BAC covering the *E(spl)-C* as described earlier for the GFPm3[9] and GFPm8 BACs[40]. Briefly, a functional *E(spl)-C* BAC[59] was modified using recombineering mediated gap-repair to introduce GFP at the N-terminus, with a GVG linker. The m3-GFP BAC was similarly engineered by replacing the ORF of m3 by those of sfGFP. Recombined regions were verified by sequencing prior to phiC31-mediated integration at the M{3xP3-RFP.attP}ZH-51D site.

To tag the endogenous m3 at its N-terminus, we used CRISPR-mediated HR. GFP knock-in was achieved by injecting three plasmids, a donor template with two gRNAs, into Cas9-expressing embryos. The gRNAs were selected using the Optimal Target Finder tool (http://flycrispr.molbio.wisc.edu/tools). Oligonucleotides were cloned into pU6-BbsI-chiRNA (Addgene #45946) as described in www.addgene.org/crispr/OConnor-Giles/ (see Supplementary Table 1 for the sequence of the gRNAs). The donor template was produced from the GFPm3 BAC. An FRT sequence was introduced 3' to the m3 gene (557 nt downstream of the stop codon of m3). The PAM sequences targeted by the gRNAs were mutated in the donor templates to avoid their Cas9-mediated cleavage. The 3xP3-RFP marker flanked by loxP sites was produced by gene synthesis and inserted 614 nucleotides downstream of the stop codon of m3. Left and right homology arms for CRISPR-mediated HR were 1 kb long. Additional cloning details will be provided upon request. A mix of donor template (200 ng/microl) and gRNA plasmids (100 ng/microl) was injected into ~500 embryos from the PBac {vas-Cas9}VK00027 stock (BL-51324). Correct recombination events were checked by PCR. Flies homozygous for the resulting knock-in *m3*[GFP-FRT] allele were viable and fertile.

CRISPR-mediated HR was used together with flp-FRT recombination to produce a small *E(spl)m7-m8* deletion. In a first step, we introduced in a single HR step two FRT sites, one 5' to the *m7* gene (172 nt 5' to the ATG of m7) and another 3' to the *m8* gene (210 nt 3' to the stop codon of m8), as well as a V5 tag at the N-terminus of the m8 protein to produce the *E(spl)*[FRTm7V5m8FRT] chromosome. To do so, the *E(spl)-C* BAC was modified in four sequential steps of recombineering-mediated gap-repair to produce a donor template with the *3xP3-RFP* marker flanked by loxP inserted just 3' to the *m8* gene (210 nt downstream of the stop codon of m8). The resulting donor was co-injected with gRNA-encoding pU6-BbsI-chiRNA plasmids (see Supplementary Table 1 for the sequence of the gRNAs) as described above. The *E(spl)*[FRTm7V5m8FRT] flies were homozygous viable. Note, however, that we were not able to detect V5-tagged m8 in the developing notum, presumably due to the low sensitivity of V5 detection. In a second step, we used flp-

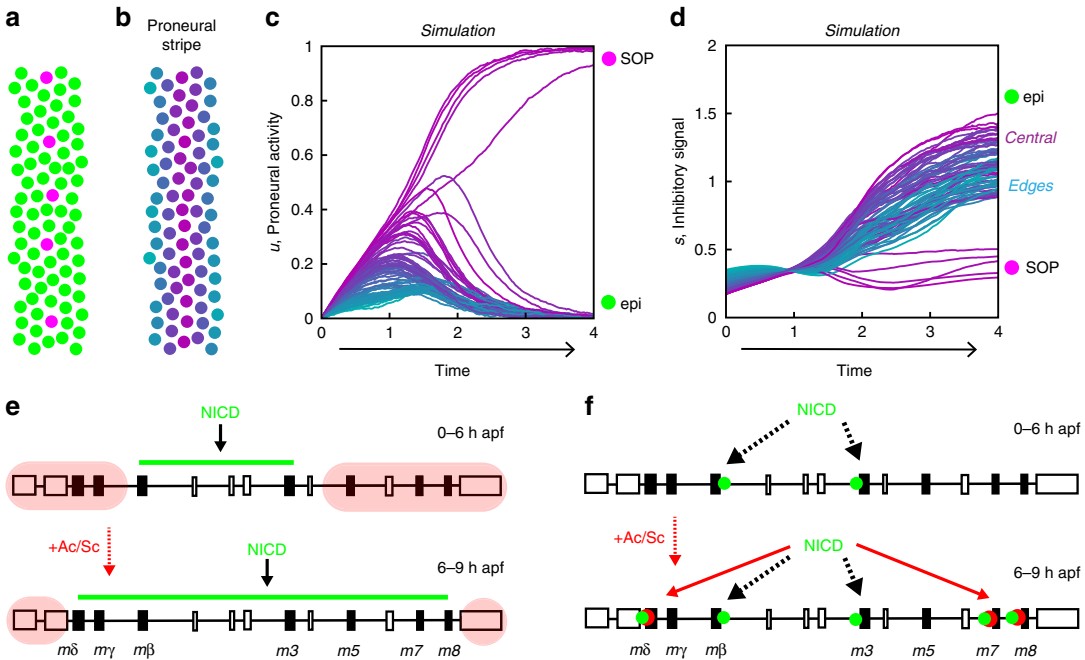

**Fig. 9** Models. **a–d** Simulation of a mathematical model for patterning in the notum[9]. Cells within a stripe eventually reach one of two stable states, representing the SOP (magenta) or epidermal (green) cell fate (**a**). Time courses of the cell state $u$, representing proneural activity (**c**), and of the inhibitory signal $s$ (**d**), which antagonizes $u$, are color coded according to their position within the stripe, as shown in **b**: cells at the center of the stripe appear in purple, whereas lateral cells are in blue. At the end of the simulation, five SOPs (magenta) have emerged at the center of the stripe (c; see their positions in **a**, whereas the other cells adopt an epidermal fate (green). Note that lateral cells (blue) are excluded first (low $u$ values in c). Also, while initial signal levels, which delimit the stripe, are highest on its sides, central cells (purple) rapidly show higher levels than their lateral counterparts (**d**). **e**, **f** Speculative models for the temporal regulation of the *E(spl)-HLH* genes. In the accessibility model (**e**), only the central region of the *E(spl)-C* is accessible for binding by CSL/NICD complexes prior to Ac and Sc expression (top; green line at 0–6 h apf; non-accessible region in red); expression of Ac and Sc renders additional binding sites accessible at 6–9 h apf (bottom). In the cooperativity model (**f**), NICD (green dots) can only bind the cis-regulatory sequences of early-onset genes which contain high-affinity CSL binding sites (top), whereas the low-affinity CSL binding sites of the late-onset genes can only be bound by NICD upon expression of Ac and Sc (red dots) through cooperative binding (bottom)

FRT recombination in the male germline using a *tub85D-flp* transgene (BL-7196) to recover a small *E(spl)m7-m8* deletion. Correct recombination was checked by PCR.

To create a null mutant allele of the *m3* gene, we selected a pair of gRNAs that would create a small deletion upon cleavage at both sites followed by non-homologous end joining (see Supplementary Table 1 for gRNA sequence). The gRNA plasmids were co-injected into PBac{vas-Cas9}VK00027 embryos. The founder flies born from injected embryos were first crossed then screened by genomic PCR (gPCR) for the presence of the expected deletion, which was detected in about a third of the founder flies ($n = 18/69$; but only 3 of the positive founders were fertile). The progeny of the 3 fertile positive founders was then screened by gPCR to isolate the corresponding deletion. Using this approach, we recovered two deletion alleles. Here, we used the *m3CR1* allele which corresponds to a 420 nucleotide (nt) deletion that removes most of the 5' UTR, the start codon and the first 88 amino acids (Supplementary Fig. 1).

To next generate a null mutant allele of the *mβ* gene onto the *m3CR1* chromosome, we followed the same approach and co-injected gRNA plasmids into M{vas-Cas9}ZH-2A;; *m3CR1* embryos (see Supplementary Table 1 for gRNA sequence). The founder flies born from injected embryos were first crossed then screened by genomic PCR (gPCR) for the presence of the expected deletion. From 62 fertile founders, we selected the three founders with the strongest positive gPCR signal and screened their progeny by gPCR. We obtained a single 531 nt deletion that removes most of the 5' UTR, the start codon and the first 148 amino acids of the *mβ* gene (Supplementary Fig. 1). This therefore produced a *m3CR1 mβCR1* double mutant chromosome.

To produce a chromosome with the *E(spl)-C* flanked by FRT sites we used the following FRT sites: (i) a first pair of FRT sites flanking the *white* gene carried by a PiggyBac element inserted in the *Nf1* gene, PBac{PB}Nf1c00617, located 13 kb proximal to the *E(spl)-C*; (ii) a second pair of FRT sites, marked by 3xP3-RFP, flanking the *m7* and *m8* genes in the *E(spl)FRTm7V5m8FRT* chromosome. These two sets of FRT sites that are distant of 56 kb were introduced onto the same chromosome by meiotic recombination, using *white+* and *RFP+* as selection marker (>5000). This locus, noted *E(spl)-CFRT0*, then underwent Flp-FRT recombination in the male germline using a *tub85D-flp* transgene and a *white−RFP+* chromosome with only two FRT sites and a *E(spl)m7-m8* deletion, noted *E(spl)-CFRT3*, was selected by gPCR. By combining *E(spl)-CFRT3* with a Flp expressed

in a tissue-specific manner, e.g., using *ap-Gal4 tub-Gal80ts UAS-flp*, we were able to efficiently delete the entire locus in a controlled manner.

Primers and cloning details can be found in Supplementary Methods.

**Flies**. The following stocks were used: GFPm3[9], GFPm8[40], RFP-Ac[9], GFP-Sc[9], *sc10-1* (BL-36541), *DlrevF10 SerRx82*, *Gbe-GFP* (pGR integrated at 51D)[60], and different deficiencies of the *E(spl)-C*: *Df(3)32.2 gro+* (a complete deletion, from P. Heitzler, CNRS, Strasbourg)[25], *Df(3)P11* (a mγ-m8 complete deletion from A. Preiss, University of Hohenheim)[39], *Df(3)X10* (a mδ-m3 deletion from A. Preiss, University of Hohenheim)[39] and *Df(3)mδ-m6*[43]. The *mib1* mutants studied here are *mib11/mib12* trans-heterozygotes[36]. Minute clones were generated using the *RpS3[Plac92]* mutation (BL-5627) with an *Ubx-flp* (BL-42720).

**Immunostainings and FISH**. Nota were dissected from staged pupae using Vannas micro-scissors, fixed in paraformaldehyde (4% in PBS 1×) and incubated in PBS 1× with 0.1% Triton X-100 and primary antibodies for 2 h at room temperature (or over-night at 4 °C), with the exception of the experiments shown in Fig. 3 for which incubation with anti-GFP was performed overnight at 37 °C in 2× SSC. The following antibodies were used: goat anti-GFP (Abcam, 1:1000), rabbit anti-DsRed (Clonetech, 1:200), goat anti-Su(H) (sc-15813, from Santa Cruz Biotechnology), mouse anti-Cut (2B10, from DSHB, 1:500), rat anti-Elav (7E8A10, from DSHB, 1:200). Secondary antibodies were from Jackson's laboratories. Following washes in PBT, nota were mounted in 4% N-propyl-galate, 80% glycerol.

FISH was performed as described in ref. [61]. Briefly, dissected tissues were fixed in 4% paraformaldehyde in PBS 1×, permeabilized in BBS 1x Triton X-100 0.5%, transferred into SSC 2× with Urea 4 M then hybridized at 37 °C in SSC 2×, Urea 4 M, Dextrane 10%, Vanydyl complex 10 mM, 0.15 mg/ml salmon sperm DNA, rinsed in SSC 2× with Urea 4 M, then whasehd in SSC 2×. The *ac* probe mRNA consisted in a set of 24 oligonucleotides (20 nt) coupled to Quasar-570 (from Stellaris Inc.). The *gfp* probe with Quasar-570 dye (VSMF-1014-5) was also from Stellaris Inc. The various *E(spl)-HLH* and *wingless* (*wg*) probes were smiFISH probes corresponding to duplexes between a set of gene-specific non-labeled primary oligonucleotides (probe set mix; equimolar mix of 23–29 different oligonucleotides with a 20 nt-long mRNA-binding moiety for a total length of

48 nt), each hybridized with the same fluorescently labeled secondary detector oligonucleotide (FLAP-X, 28 nt-long) coupled to Cy3 or Cy5[62]. smiFISH probes were prepared as described in ref. [62]. Briefly, FLAP-X oligonucleotides were annealed with the probe set mix in Tris-HCl 50 mM pH = 7.5, NaCl 100 mM, MgCl2 10 mM using a thermocycler (85 °C, 3 min; 65 °C, 3 min; 25 °C 3 min). For dual color FLAP-X, the hetero-duplexes corresponding to the C3-coupled and Cy5-coupled FLAP probes were mixed at the hybridization step. For double immuno-FISH stainings, primary antibodies were incubated together with the smiFISH probes in the hybridization mix. Secondary antibodies were incubated in PBS 1× with 0.1% Triton X100 after the hybridization was step. Both labeled and non-labeled oligonucleotides were obtained from IDT Inc.

**Microscopy and image analysis**. Images were acquired using a confocal Zeiss LSM780 microscope with ×63(PL APO, N.A. 1.4 DIC M27) and ×40 (PL APO, N. A. 1.32 DIC M27) objectives. Adult flies were imaged using a Zeiss Discovery V20 stereo-macroscope using a ×1.0 (PlanApo S FWD 60 mm) objective.

Gene expression profiles were computed as follows. First, images were segmented and reporter levels quantified as described in[9], with a manual correction step to exclude objects that were not nuclei (e.g., signal from the cuticle). Reporter levels were then normalized such the maximal level of each reporter in each image was of order one, and nuclei were defined to exhibit detectable expression of a reporter if its normalized level exceeded a set threshold (0.2). Images were manually annotated to identify the centerline of stripe 3, and after binning the nuclei according to their position relative to this line, gene expression profiles were obtained by summing the intensity (minus the detection threshold) of the nuclei in each bin.

**Western blot**. Brain-discs complexes were dissected from third instar and protein extracts were prepared in Tris 50 mM pH8, NaCl150mM, SDS 1% buffer. Protein concentration was determined using a Thermo Scientific NanoDrop1000 Spectrophotometer and 100 μg of protein was loaded per lane on 4–20% precast Miniprotean TGX gels (Biorad) for SDS-PAGE. Proteins were transferred onto 0.2 μm Nitrocellulose membranes (Biorad). An HRP-coupled anti-GFP (Abcam #6663, 1:2000) was used to detect the GFP and GFPm3 proteins using SuperSignal WestFemto (Thermo Scientific)

**Mathematical model**. The model shown in Fig. 9a–d[9] describes the state of each cell (labeled by an index $i$) by a scalar variable $u_i$, which varies in time according to

$$\tau \frac{du_i}{dt} = f(u_i - s_i) - u_i + \eta_i(t), \qquad (1)$$

where $\tau$ is a characteristic time, $f$ is a sigmoidal function that varies between 0 and 1, $s_i$ is the signal received by the cell, and $\eta_i(t)$ is a stochastic term that allows for fluctuations in the dynamics. The term $f(u_i - s_i)$ represents the balance between self-activation and inhibitory signaling, with the signal $s_i$ combining a time-dependent extrinsic signaling gradient, and the signal received from neighboring cells,

$$s_i = s_0(x_i, t) + \sum_{j \neq i} c_{ij} D * (u_j). \qquad (2)$$

In this equation, $s_0(x_i,t)$ describes the extrinsic signaling gradient as a function of the cell's position $x_i$ along the medial-distal axis and time, the coefficient $c_{ij}$ is a decaying function of the distance between cells $i$ and $j$, representing the range of signaling, and $D*(u)$ is the level of active Delta ligand produced by a cell according to its state $u$ (signaling is instantaneous). The model is simulated with periodic boundary conditions and fixed cell positions, with a disordered cell arrangement generated using a vertex model for epithelial tissue dynamics[63]. Here, the extrinsic gradient represents the gradual transition between early, proneural-independent Notch activity, providing a negative template for stripe 3, and the inhibitory signal produced by stripes 1 and 5, and the model describes the sequential emergence of stripes 3, then 2 and 4 (see ref. [9] for details).

**Reporting summary**. Further information on research design is available in the Nature Research Reporting Summary linked to this article.

## Data availability
The authors declare that all data supporting the findings of this study are available within the article and its supplementary information files or from the corresponding author upon reasonable request.

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

## Acknowledgements

We thank H. Bellen, P. Heitzler, A. Preiss, the Bloomington *Drosophila* Stock Center, the Kyoto *Drosophila* Genetic Research Center, the Developmental Studies Hybridoma Bank (DSHB), and Flybase for flies, antibodies and other resources. We thank V. Roca for embryo injection and M. Reyngoud for technical help. We thank L. Bally-Cuif, F. Bernard, and D. Henrique for critical reading and all lab members for discussion. This work was funded by ANR-10-LABX-0073 and ANR-16-CE13-0003 grants.

## Author contributions

L.C. performed all experiments with the exception of the molecular biology experiments performed by K.M.; F.C. and F.S. designed the study, analyzed the data, and wrote the paper.

## Additional information

**Competing interests:** The authors declare no competing interests.

