## [Peer Review File · Nature Communications]

Reviewers' Comments:

Reviewer #1:

Remarks to the Author:

Review to: Regulation of specific Enhancer of split-HLH genes by proneural factors shapes Notch output dynamics during bristle patterning in *Drosophila*. Courturier et al.

The manuscript deals with the E(spl)-C in *Drosophila*. The complex consists of genes encoding bearded like proteins and seven very similar bHLH proteins. The E(spl)-HLH genes are partly redundant in their function, since single gene mutations are without detectable phenotype. Moreover, the genes are expressed in a similar but not identical pattern in imaginal disks. In this manuscript, the complex was studied in more detail and the authors come to the conclusion that the different bHLH proteins may have distinct function during bristle development in the fly. The authors show that the $m\beta$ and $m3$ genes are expressed early on to define the regions where proneural stripes may form. These two genes can be replaced functionally by $m\delta$. After formation of the proneural stripes the three HLH-genes $m7$, $m8$ and $m\delta$ become expressed in low and intermediate Sc and Ac expressing cells to select SOP fate by lateral inhibition. The authors support their findings with a mathematical model.

The manuscript is well written; the experiments are technically sound and overall support the authors' conclusions. However, there are several points, which should be addressed by the authors prior to acceptance.

Major:

1. On several occasions, data are 'not shown'. Please provide the missing data in a supplemental file.

It may be acceptable not to show the $m3$ wild type pattern (p8), but it is not in the case of the 'small amorphic wing' phenotype of the E(spl)-C deletion (p9). Likewise, the normal phenotype of the homozygous or hemizygous $m3/m\beta$ double mutants (p11) should be presented, as well as the GFP $m8$ data (p14), GFP $m\beta$ and GFP $m7$ (p15, Fig 7) data.

2. The erratic use of either GFP $m8$ or GFP $m7$ is disturbing. Please be consistent and use either the one or the other. Moreover, provide evidence in a supplemental file that the two are indeed identical in their expression profile.

3. Fig 5 E-H: which probe was used? Ac or Sc? There is a discrepancy between the figure and the legend.

The labelling with this probe is extremely weak and very hard to see. At present, these pictures hardly support the authors' claims. For example, I am unable to see any labeling underneath or next to the asterisks in Fig 5 E,F. Please improve the quality of these pictures! For example, the authors may inverse the colors and/or use different coloring.

4. BAC transgenes – CRISPR-based genome engineering. In the materials and methods, the authors refer the reader to 'primers and cloning details available upon request'. While as a reader, I may not be interested in these details, as a reviewer I have to be. Please provide these details for reviewing purposes.

5. Page 10 - protein stability. The authors claim that replacement of $m3$ by GFP gives stronger signals and a broader expression pattern than the $m3$ GFP fusion. The authors conclude that the $m3$ GFP fusion protein turns over more rapidly than GFP. The data shown do not allow drawing this conclusion since they are based on the apparent fluorescent intensity of different specimens treated with antibodies. Firstly, the authors cannot exclude a different affinity of the antibodies to GFP versus fusion protein. Secondly, handling could have differed, as well as the developmental age of the specimens. Thirdly, expression levels may differ, for example due to secondary structure of the transcripts.

Please provide qRT-PCR data and a quantitative Western blot analysis.

Please provide a legend to Fig 3 G. What exactly are the differences between GFP- $m3$ and $m3$ -GFP?

Minor:

- Reference 2 is not yet published – please omit
- On page 12 (second paragraph), please refer to Fig 5F: Derepression of the *mδ* gene.
- Discussion is a bit lengthy; and the evolutionary aspects are somewhat superficial. For example: Mosquitos have a single E(spl)-HLH gene with conserved CSL and proneural regulatory sites. As proposed, the gene may indeed be controlled by both, proneural proteins as well as Notch activity. Still the mechanisms underlying bristle patterning must be different from *Drosophila* (as seen by the outcome), since with a regulatory network between E(spl)-HLH genes is hardly envisaged with a single gene copy (i.e. cross-repression or –activation). Moreover, whereas the architecture of regulatory sequences appears to be conserved, their function - to my knowledge - has only be experimentally addressed for the bearded orthologs

Reviewer #2:

Remarks to the Author:

Review of: “Regulation of specific Enhancer of split-HLH genes by proneural factors shapes Notch output dynamics during bristle patterning in *Drosophila*” by Couturier et al.

Summary: In this paper by Couturier et al, the authors perform a thorough investigation of the roles of E(spl) transcription factors in mediating Notch fate decisions during microchaetae sensory bristle formation in *Drosophila*. The work builds upon a recent Science paper from the Schweisguth lab that showed that the Notch signal lies both up- and downstream of the Ac/Sc proneural genes during microchaetae development as opposed to just downstream during macrochaetae development. The major findings in this study are the demonstration that distinct subsets of E(spl) genes respond to Notch signaling during pre-patterning (i.e. establishing expression rows of Ac/Sc positive cells) as compared to during SOP fate selection and cell type specification downstream of Ac/Sc. By and large the experiments are well done and the paper is very well written. In addition, the numerous fly lines generated will be of interest to those studying Notch signaling in flies. However, the overall conclusions are largely expected. It has been known for a considerable time that the E(spl) locus contains a large number of bHLH factors that respond to Notch signals in only a subset of tissues. Moreover, it has also been previously shown by Caudy and Posakony that the Ac/Sc genes can synergize with the Notch pathway to induce specific E(spl) genes during SOP selection and cell fate specification. Thus, while the very large body of work is thorough, elegant, and convincing, it also does not significantly alter or advance the way we think about how the Notch, E(spl)/Hes, and Ac/Sc genes affect cell fate specification. Hence, my major concern is whether this paper represents a significant enough advance to warrant publication in *Nature Communications*. In addition, I do have several relatively minor comments/suggestions that the authors can consider prior to publication.

Additional concerns/comments:

1) The experiments in Figure 2 remove the entire E(spl) locus and observe a neurogenic phenotype reminiscent of Notch mutant flies. The logical interpretation is that without the E(spl) target genes the Notch gene regulatory network is not appropriately controlled. But it is unclear if the authors know that the Notch signal is still active in the absence of E(spl) genes. Are other Notch target genes such as the NRE still active in these cells? Is Delta/Notch expression normal, etc.

2) A minor comment: In a supplemental figure, it would be helpful to show exactly what part of the coding regions are deleted in m3 in the m3CR1 allele. The authors stated that the first 88 amino acids are removed – but does that take out the bHLH domain? Is there any remaining ATG that could be used to make a truncated protein in-frame with the bHLH if it was not deleted?

3) On page 12, the authors wrote: “In contrast, no significant up-regulation was detected for the *mg*, *m5*, *m7* and *m8* genes in the DC region of mutant pupae (Fig 5G,H).” At least in the picture in the pdf file, it appears to me that *mg*, *m5*, *m7* and *m8* are considerably stronger in the mutant

animal in 5H than the wild type animal in 5G. It would be much more convincing if the authors quantify the changes or lack of changes in mRNA expression in these animals.

4) For the stains in Fig 6L, it is really hard to tell which cells are RFP-AC positive/negative and which cells are GFPm7 positive/negative. It appears that many of the cells that are GFP-positive are RFP negative. It would be much better to show the isolated channels in black and white. Also, there are no statistics/standard deviation of the quantitation on the expression levels in panels J and M. How consistent is this pattern between animals?

5) For the clonal analysis in Fig 7e,f, I think it is again imperative to show the isolated channels in black and white. This would make it much easier to interpret the data.

Reviewer #3:

Remarks to the Author:

Manuscript NCOMMS-18-38047 entitled "Regulation of specific Enhancer of split-HLH genes by proneural factors shapes Notch output dynamics during bristle patterning in *Drosophila*" by Couturier et al.

In this manuscript Couturier et al. dissect the function of the genes of the Enhancer of split complex (E(spl)-C) during the development of the small bristle sensilla of *Drosophila*. It is a follow up of their interesting investigation of the role of the Notch pathway during the patterning of the five stripes of the mechano-sensory bristles termed microchaete which cover the notum of *Drosophila melanogaster*.

In this manuscript the investigate the function of the main target during this patterning process, the E(spl)-C. They found that the genes of the complex can be divided into an early and later acting group which have different functions. The first acting group dependent only on Notch signaling set up the placing of the equivalence groups, which is defined by the expression of the proneural factor Achaete. The later acting group mediates the Notch mediated selection of the neural precursor within the equivalence group.

The reported results are excellent and give a deeper insight of the function of the E(spl) complex, which are targets of the Notch signaling in many metazoans. This insight was so far missing. The results are supported by excellent experiments, which use state of the art techniques. In addition, the generated tools will be of interest for many researchers in the field of Notch signaling. The manuscript is perfectly suited for publication in Nature communications and I strongly recommend the publication of the manuscript.

However, a few points should be considered:

-I was surprised to read that the used *mib1* alleles constitute a strong hypomorphic situation, although the Schweisguth group described them as null alleles before. If there is new evidence then this should be provided.

-In the discussion the recent paper of the Posakony and Klein groups about the selection of the neural precursor of the macrochaete should also be discussed in the light of the here presented new results. In many pictures of this manuscript it is clearly visible that the development of this type of the bristle is also affected and the selection within the proneural cluster occurs in both developmental situations.

-Also, it should be made clear that the expression of the E(spl) genes are not necessarily reflecting the strength of Notch activity. For this a Notch activity reporter needs to be shown and compared with the expression of the E(spl) genes.

In summary, I strongly recommend the acceptance of this very nice work for publication after taking the mentioned points into account.

Reviewer #1

*The manuscript deals with the $E(spl)$ -C in *Drosophila*. The complex consists of genes encoding bearded like proteins and seven very similar bHLH proteins. The $E(spl)$ -HLH genes are partly redundant in their function, since single gene mutations are without detectable phenotype. Moreover, the genes are expressed in a similar but not identical pattern in imaginal disks.*

*In this manuscript, the complex was studied in more detail and the authors come to the conclusion that the different bHLH proteins may have distinct function during bristle development in the fly. The authors show that the $m\delta$ and $m3$ genes are expressed early on to define the regions where proneural stripes may form. These two genes can be replaced functionally by $m\delta$. After formation of the proneural stripes the three HLH-genes $m7$, $m8$ and $m\delta$ become expressed in low and intermediate *Sc* and *Ac* expressing cells to select SOP fate by lateral inhibition. The authors support their findings with a mathematical model.*

The manuscript is well written; the experiments are technically sound and overall support the authors' conclusions. However, there are several points, which should be addressed by the authors prior to acceptance.

We thank the referee for her/his generally positive assessment of our work and address her/his specific points below:

Major:

1. On several occasions, data are 'not shown'. Please provide the missing data in a supplemental file. It may be acceptable not to show the $m3$ wild type pattern (p8), but it is not in the case of the 'small amorphic wing' phenotype of the $E(spl)$ -C deletion (p9). Likewise, the normal phenotype of the homozygous or hemizygous $m3/m\delta$ double mutants (p11) should be presented, as well as the GFPm8 data (p14), GFPm δ and GFPm7 (p15, Fig 7) data.

We now present these data in the following supplemental figures:

- $E(spl)$ -C wing phenotype: we agree with the referee that our description of the $E(spl)$ -C wing phenotype was incomplete and needed documentation. To illustrate the range of observed phenotypes (small-sized wings, vein thickening), two individuals are shown in **Fig S1d,e**, with control. Please note that the wing margin remains properly specified upon loss of $E(spl)$ -C activity, i.e. wings are actually not amorphic (we realized that the ones we observed as amorphic were actually simply folded).
- bristle pattern in $m3$ and $m3 m\delta$ mutant flies: now shown in **Fig S1a',b'**.
- the expression of GFPm8 in stripe 3 is now documented in **Fig 6n-o''**. A graph showing the profile of GFPm8 accumulation relative to RFP-Ac is shown in **Fig 6p**. We further show in **Fig S3** that GFPm7 (protein) is co-expressed with $m8$ (mRNA) and, conversely, that GFPm8 (protein) is co-expressed with $m8$ (mRNA).
- the proneural-dependent expression of GFPm β and GFPm8 are now shown in **Fig S4a-d'** (with GFPm7 now shown in **Fig 7c,d**).

2. The erratic use of either GFPm8 or GFPm7 is disturbing. Please be consistent and use either the one or the other. Moreover, provide evidence in a supplemental file that the two are indeed identical in their expression profile.

As mentioned above, we now provide the data for both GFPm7 and GFPm8 and have re-arranged the figures to improve consistency and clarity: i) **Fig 6** now shows both GFPm7 and

GFPm8 data; ii) the proneural-dependency data have now been re-organized between **Fig 7** (GFPm7 only) and **Fig S4** (GFPm8 only). Additionally, to directly compare their expression patterns, we have examined the expression of endogenous m7 (by FISH) and GFPm8 (by anti-GFP staining), and reciprocally. The results shown in **Fig S3** clearly indicate that these two genes are expressed similarly. Note, however, that m7 seems to be expressed earlier and at higher levels than m8.

3. Fig 5 E-H: which probe was used? Ac or Sc? There is a discrepancy between the figure and the legend.

The labelling with this probe is extremely weak and very hard to see. At present, these pictures hardly support the authors' claims. For example, I am unable to see any labeling underneath or next to the asterisks in Fig 5 E,F. Please improve the quality of these pictures! For example, the authors may invert the colors and/or use different coloring.

We apologize for the typo, the *ac* probe was used in the previous version of this paper. However, since this probe gave a weak signal, we repeated these experiments using wingless (*wg*) as a position landmark (*wg* is expressed along proneural stripe 5). These new data are now in **Fig 5e-h'**. To better see the expression of the *E(spl)*-*HLH* genes, the corresponding FISH signal is now shown in separate panels.

4. BAC transgenes – CRISPR-based genome engineering. In the materials and methods, the authors refer the reader to 'primers and cloning details available upon request'. While as a reader, I may not be interested in these details, as a reviewer I have to be. Please provide these details for reviewing purposes.

The information about primers and cloning details is now provided as a new **Supplemental Materials and Methods** file.

5. Page 10 - protein stability. The authors claim that replacement of m3 by GFP gives stronger signals and a broader expression pattern than the m3GFP fusion. The authors conclude that the m3GFP fusion protein turns over more rapidly than GFP. The data shown do not allow drawing this conclusion since they are based on the apparent fluorescent intensity of different specimens treated with antibodies. Firstly, the authors cannot exclude a different affinity of the antibodies to GFP versus fusion protein. Secondly, handling could have differed, as well as the developmental age of the specimens. Thirdly, expression levels may differ, for example due to secondary structure of the transcripts.

Please provide qRT-PCR data and a quantitative Western blot analysis.

Please provide a legend to Fig 3 G. What exactly are the differences between GFP-m3 and m3-GFP?

The referee may have (understandably) got confused with our nomenclature (suggestions for a clearer nomenclature are actually welcome!). The following nomenclature was used:

1. **GFPm3**: this is N-terminal GFP fused to m3, both in the GFPm3 BAC and in the $m3^{GFP-FRT}$ KI allele, i.e. the KI allele and the BAC transgene encode the same fusion protein
2. **m3-GFP**: this is a transcriptional reporter, with GFP replacing the ORF of m3, i.e. GFP is expressed under the regulatory sequence of m3 in the BAC transgene

The 'm3GFP' and 'GFP-m3' nomenclature used above by the referee was not present in our paper (unless we missed a typo).

Whatever this nomenclature issue, the referee questions whether the difference in signal intensity (anti-GFP staining) seen between GFPm3 and m3-GFP might result from:

- affinity difference of anti-GFP on GFP alone (produced by m3-GFP) versus GFPm3: this seems unlikely given that GFP is properly folded in the fusion protein GFPm3 (direct GFP fluorescence) but hard to rule without measuring affinity. As suggested by the referee, we performed WB (under denaturing conditions; hence, the affinity issue becomes irrelevant) and compared the relative accumulation levels of GFP (produced by the m3-GFP transgene) and GFPm3 (produced by the GFPm3 BAC; both inserted at the same landing site). In extracts prepared from brain-discs complexes dissected from third instar larvae, we were able to detect GFP (from m3-GFP) but not the GFPm3 fusion protein (**Fig S2c**). This result supports our conclusion that the GFPm3 protein is less abundant than GFP in this tissue.

- differences in handling and/or staging: we can rule out this possibility since this difference was seen in multiple wing discs which were processed in parallel. Also, the extracts used for WB were prepared from >10 larvae, hence averaging differences in inter-individual staging (**Fig S2c**).

- differences in mRNA structure, hence mRNA stability, translation and levels of GFP: the fact that the 5' and 3' UTR of the *m3* gene were identically maintained in the two construct makes this interpretation unlikely. Nevertheless, to address this possibility, we have tested whether the observed difference in protein levels correlates with a difference in mRNA levels. We compared the relative levels of proteins (GFPm3 or GFP proteins, using anti-GFP) and mRNAs (using a *gfp* FISH probe) and found a much stronger anti-GFP signal in m3-GFP pupae relative to GFPm3 pupae, whereas the FISH signal appeared similar (**Fig S2 d-e**). While the referee suggested to perform RT-qPCR, we feel that FISH may be more informative as it provides spatial information. Since translation most often correlates with mRNA stability, the simplest interpretation of our data is that GFPm3 is less stable than GFP.

We now conclude: ' This (difference) appears to result from the time integration of the Notch input at the protein level rather than at the RNA level (Fig S2), suggesting that GFPm3 turns over more rapidly than GFP.'

Minor:

- Reference 2 is not yet published – please omit

As this review is still in revision, we have deleted this reference.

*- On page 12 (second paragraph), please refer to Fig 5F: Derepression of the *mδ* gene.*

done

- Discussion is a bit lengthy; and the evolutionary aspects are somewhat superficial. For example:

*Mosquitos have a single *E(spl)*-HLH gene with conserved CSL and proneural regulatory sites.*

As proposed, the gene may indeed be controlled by both, proneural proteins as well as Notch

activity. Still the mechanisms underlying bristle patterning must be different from Drosophila (as seen by the outcome), since with a regulatory network between E(spl)-HLH genes is hardly envisaged with a single gene copy (i.e. cross-repression or –activation). Moreover, whereas the architecture of regulatory sequences appears to be conserved, their function - to my knowledge - has only be experimentally addressed for the bearded orthologs

We agree that the mosquito context deserves further experimental analysis. We have followed the referee's advice and no longer refer to this context in the discussion.

Reviewer #2

Summary: In this paper by Couturier et al, the authors perform a thorough investigation of the roles of E(spl) transcription factors in mediating Notch fate decisions during microchaetae sensory bristle formation in Drosophila. The work builds upon a recent Science paper from the Schweisguth lab that showed that the Notch signal lies both up- and downstream of the Ac/Sc proneural genes during microchaetae development as opposed to just downstream during macrochaetae development. The major findings in this study are the demonstration that distinct subsets of E(spl) genes respond to Notch signaling during pre patterning (i.e. establishing expression rows of Ac/Sc positive cells) as compared to during SOP fate selection and cell type specification downstream of Ac/Sc. By and large the experiments are well done and the paper is very well written. In addition, the numerous fly lines generated will be of interest to those studying Notch signaling in flies. However, the overall conclusions are largely expected. It has been known for a considerable time that the E(spl) locus contains a large number of bHLH factors that respond to Notch signals in only a subset of tissues. Moreover, it has also been previously shown by Caudy and Posakony that the Ac/Sc genes can synergize with the Notch pathway to induce specific E(spl) genes during SOP selection and cell fate specification. Thus, while the very large body of work is thorough, elegant, and convincing, it also does not significantly alter or advance the way we think about how the Notch, E(spl)/Hes, and Ac/Sc genes affect cell fate specification. Hence, my major concern is whether this paper represents a significant enough advance to warrant publication in Nature Communications. In addition, I do have several relatively minor comments/suggestions that the authors can consider prior to publication.

We thank the referee for her/his generally positive assessment of our work. While the referee may have predicted the patterning dynamics reported herein for the different E(spl)-HLH genes, it had not been reported nor proposed. The published literature on the regulation of the E(spl)-HLH genes by proneural factors, including the work by Caudy and Posakony which we cite, makes no mention of the implication of the reported synergy for patterning dynamics, which we study here.

Additional concerns/comments:

1) The experiments in Figure 2 remove the entire E(spl) locus and observe a neurogenic phenotype reminiscent of Notch mutant flies. The logical interpretation is that without the E(spl) target genes the Notch gene regulatory network is not appropriately controlled. But it is unclear if the authors know that the Notch signal is still active in the absence of E(spl) genes. Are other Notch target genes such as the NRE still active in these cells? Is Delta/Notch expression normal, etc.

We have now examined the activity of a generic Notch activity reporter. Gbe-GFP (from S. Bray, Cambridge) was chosen over NRE-GFP (from G. Merdes, Basel) as it gave a stronger signal (see Furiols and Bray, Current Biology, 2001 for discussion). Expectedly, Gbe-GFP expression differs from those of each of our E(spl)-HLH reporters considered individually (**Fig S2a**). Also, in a thought experiment, it probably differs from the combination of all seven E(spl)-HLH. This can be easily explained by the context-specific response of the E(spl)-HLH genes to Notch, while the Notch-dependent expression of Gbe-GFP is generally assumed to reflect a generic response. Interestingly, we observed that the Notch signaling activity reported by GFPm3 in the notum region at 2h apf can hardly be detected by Gbe-GFP (**Fig S2b**). Thus, it is possible that the Gbe-GFP reported is also context-specific and/or that our GFPm3 reporter is more sensitive than Gbe-GFP to a low Notch signal in this tissue.

Lastly, we have experimentally addressed whether the Notch signal is still active in the absence of *E(spl)-C* genes. Specifically, we found that deletion of the *E(spl)-C* did not affect the expression of Delta as shown in Fig R1 below for in a conditional tissue-specific loss of the *E(spl)-C* in *Df(3)P11 / E(spl)^{FRT3}* pupae. Also, expectedly, increased Notch signaling activity (monitored using Gbe-GFP) was seen within proneural clusters (but not outside) upon deletion of the *E(spl)-C* (see Fig R1 below). This effect suggests that increased proneural activity (in the absence of its *E(spl)*-HLH repressors) results in high Delta and Neuralized expression in these clusters. These results are not unexpected. We therefore provide them here for the referees but it may not be necessary to document these findings in the paper.

Fig R1: Notch signaling in *E(spl)-C* mutant notum

No significant change in the expression of the synthetic Notch reporter GbeGFP (green) was seen upon conditional tissue-specific deletion of the *E(spl)-C* in the pupal notum (0h apf), with the notable exception of the proneural clusters where increased levels of Gbe-GFP expression (Notch signaling) were seen (see DC cluster, at the base of stripe 5). Similarly, the expression of Delta (red) remained largely unchanged (see the two bands of Delta expression over stripes 1 and 5).

genotypes

control: *M[3xP3-RFP, NRE-pGR]51D / +* (1 copy of Gbe-GFP)

conditional deletion: *Ubx-flp / + ; M[3xP3-RFP, NRE-pGR]51D / + ; Df(3)P11 / E(spl)-C^{FRT3}*

2) A minor comment: In a supplemental figure, it would be helpful to show exactly what part of the coding regions are deleted in *m3* in the *m3CR1* allele. The authors stated that the first 88 amino acids are removed – but does that take out the bHLH domain? Is there any remaining ATG that could be used to make a truncated protein in-frame with the bHLH if it was not deleted?

This information is now provided in **Fig S1a,b**. In brief, the deletions in the *m3* and *mβ* genes remove the bHLH and Orange domains.

3) On page 12, the authors wrote: “In contrast, no significant up-regulation was detected for the *mg*, *m5*, *m7* and *m8* genes in the DC region of mutant pupae (Fig 5G,H).” At least in the picture in the pdf file, it appears to me that *mg*, *m5*, *m7* and *m8* are considerably stronger in the mutant animal in 5H than the wild type animal in 5G. It would be much more convincing if the authors quantify the changes or lack of changes in mRNA expression in these animals.

The referee is correct, i.e. at least one of these five genes is indeed significantly up-regulated in the hinge and notum region distal to the DC region in double mutant pupae. This is again indicative of negative cross-regulation between E(spl)-HLH factors. We did not comment on this observation earlier because our focus was on the notum region, where no such up-regulation is detected. We understand that this understatement may actually be confusing. In this revised version, we have now repeated these experiments using *wg* as a marker (**Fig 5e-h'**; see our reply to referee #1). We confirmed ectopic *mδ* expression in the DC region (as well as in more distal region, close to the hinge) and also observed again the ectopic expression of some other E(spl)-HLH genes in this distal region (noted by the referee). This de-repression is now noted in the text.

We did not quantify this change in expression level because the difference can be clearly observed and because it is not clear that FISH is quantitative for these genes in this tissue.

4) For the stains in Fig 6L, it is really hard to tell which cells are RFP-AC positive/negative and which cells are GFPm7 positive/negative. It appears that many of the cells that are GFP-positive are RFP negative. It would be much better to show the isolated channels in black and white. Also, there are no statistics/standard deviation of the quantitation on the expression levels in panels J and M. How consistent is this pattern between animals?

We now show separate channels (**Fig 6i-o''**), and also provide the standard errors by the mean (**Fig 6j,m,p**). Despite the inherent variability in Ac and E(spl) expression, timing and stainings, the essential, qualitative features of the patterns, whereby *m3* is expressed mostly in the interstripes, and *m7/m8* within the stripes, are consistent among pupae.

5) For the clonal analysis in Fig 7e,f, I think it is again imperative to show the isolated channels in black and white. This would make it much easier to interpret the data.

We now show GFPm3 and GFPm7 in separate channels (**Fig 7e-f'**)

Reviewer #3

In this manuscript Couturier et al. dissect the function of the genes of the Enhancer of split complex (E(spl)-C) during the development of the small bristle sensilla of Drosophila. It is a follow up of their interesting investigation of the role of the Notch pathway during the patterning of the five stripes of the mechano-sensory bristles termed microchaete which cover the notum of Drosophila melanogaster.

In this manuscript they investigate the function of the main target during this patterning process, the E(spl)-C. They found that the genes of the complex can be divided into an early and later acting group which have different functions. The first acting group dependent only on Notch signaling set up the placing of the equivalence groups, which is defined by the expression of the proneural factor Achaete. The later acting group mediates the Notch mediated selection of the neural precursor within the equivalence group.

The reported results are excellent and give a deeper insight of the function of the E(spl) complex, which are targets of the Notch signaling in many metazoans. This insight was so far missing. The results are supported by excellent experiments, which use state of the art techniques. In addition, the generated tools will be of interest for many researchers in the field of Notch signaling. The manuscript is perfectly suited for publication in Nature communications and I strongly recommend the publication of the manuscript.

We thank the referee for her/his very positive and supportive comments.

However, a few points should be considered:

-I was surprised to read that the used mib1 alleles constitute a strong hypomorphic situation, although the Schweisguth group described them as null alleles before. If there is new evidence then this should be provided.

We thank the referee for pointing out this inconsistency. Indeed, we reported in our 2005 PLoS Biology paper that the mib1(1) allele (noted D-mib1 both in this earlier paper and below) behaves as a genetic null allele. Specifically, we stated that the 'D-mib1, D-mib2, and D-mib3 alleles appear to be genetically null alleles since the phenotypes of D-mib1/D-mib1 and D-mib1/D-mib3 mutant pupae are indistinguishable from the ones seen in D-mib1/D-mib2 and D-mib2/D-mib3 pupae.' However, we also reported in that paper that the mib1(1) allele is not a molecularly null allele since it corresponds to the insertion of transposable element leaving intact the structure of mib1 mRNA. This therefore leaves open the possibility that low levels of wild-type (functional) Mib1 protein is produced by this mutant allele.

In light of this, we now conservatively state that '*trans*-heterozygote *mib1¹/mib1²* may correspond to a strong hypomorphic condition'. If deemed relevant, we can explain the underlying rationale of this statement in the Methods section.

-In the discussion the recent paper of the Posakony and Klein groups about the selection of the neural precursor of the macrochaete should also be discussed in the light of the here presented new results. In many pictures of this manuscript it is clearly visible that the development of this type of the bristle is also affected and the selection within the proneural cluster occurs in both developmental situations.

We thank the referee for this suggestion. Indeed, we believe that the model proposed here is of general significance and should apply to the selection of the SOPs producing macrochaetes. For instance, our data clearly suggest that the *m7* and *m δ* genes are expressed late in the DC proneural clusters - in addition to *m3* and *m β* (**Figs 4 and 5e'**). However, to avoid redundancy, we did not experimentally study the determination of macrochaete SOP here. As suggested, we simply mention this point in the discussion and cite these two papers in that context (Castro et al. 2005; Troost et al., 2015).

-Also, it should be made clear that the expression of the E(spl) genes are not necessarily reflecting the strength of Notch activity. For this a Notch activity reporter needs to be shown and compared with the expression of the E(spl) genes.

We can only agree! Our study is one further illustration that the expression of the E(spl) genes does not directly reflect the strength of Notch signaling (NICD levels): some genes are expressed while others are not (see early patterning), or are expressed at the same time but not in the same cells (see *m3* vs *m7* during stripe attrition and SOP resolution). We also show here that some of these genes, e.g. *m8* (which is sometime used as a Notch activity reporter; see *m8-lacZ*), can even be expressed in the absence of Notch signaling (see our analysis of the *DI Ser* mutant clones). To make it clear, we now state:

'Consistent with the notion that these Notch targets are expressed in a context-specific manner^{25, 26, 28, 29}, 'the expression of the *E(spl)*-*HLH* genes are not necessarily reflecting the strength of Notch activity, and their individual or combined expression may not perfectly match the pattern of Notch signaling, which might be revealed using a synthetic NICD-responsive reporter⁴² (**Fig S2a,b**).'

Also, we have studied the expression of a classic Notch activity reporter, *Gbe*-GFP, which responds to nuclear NICD, and compared its expression with those of our reporters (**Fig S2a,b**). As indicated above, the pattern of *Gbe*-GFP did not match the pattern of any of our GFP-E(spl) reporters and does not seem to match (in a thought experiment) the combined pattern of these GFP-tagged proteins. At least two factors likely contribute to these differences. First, the E(spl)-HLH genes are regulated by other factors, which contribute to their context-specific expression. Second, because it is a transcriptional reporter, *Gbe*-GFP may fail to report the dynamics of Notch signaling, due to the stability of the GFP (as shown here for *m3*-GFP). However, our observation that *Gbe*-GFP was not detectably expressed in the two stripes of cells which express both *m3* and *m β* in the notum of 0-2h *apf* pupae is more difficult to rationalize (see **Fig S2a,b**).

Reviewers' Comments:

Reviewer #1:

Remarks to the Author:

The authors have changed their manuscript according my suggestions, therefore I support publishing of the manuscript. However, let me note one more point which the authors should address prior to publication: please add scale bars in your figures.

Reviewer #2:

Remarks to the Author:

The authors did a great job of addressing all of my experimental concerns. Moreover, in re-reading the revised paper, I believe the overall quality and impact of their findings are suitable for publication in Nature Communications. Hence, I am highly supportive of acceptance.

Reviewer #3:

Remarks to the Author:

The revised manuscript has fulfilled many of my request. As I favored already its publication, I recommend its publications now.